# Erythromycin Formulations—A Journey to Advanced Drug Delivery

**DOI:** 10.3390/pharmaceutics14102180

**Published:** 2022-10-13

**Authors:** Vera-Maria Platon, Brindusa Dragoi, Luminita Marin

**Affiliations:** 1“Petru Poni” Institute of Macromolecular Chemistry of Romanian Academy, 700487 Iasi, Romania; 2TRANSCEND Research Center, Regional Institute of Oncology, 700483 Iasi, Romania

**Keywords:** erythromycin, liposomes, solid–lipid nanoparticles, mesoporous oxides, metallic-based nanoparticles

## Abstract

Erythromycin (ERY) is a macrolide compound with a broad antimicrobial spectrum which is currently being used to treat a large number of bacterial infections affecting the skin, respiratory tract, intestines, bones and other systems, proving great value from a clinical point of view. It became popular immediately after its discovery in 1952, due to its therapeutic effect against pathogens resistant to other drugs. Despite this major advantage, ERY exhibits several drawbacks, raising serious clinical challenges. Among them, the very low solubility in water and instability under acidic conditions cause a limited efficacy and bioavailability. Apart from this, higher doses promote drug resistance and undesirable effects. In order to overcome these disadvantages, during the past decades, a large variety of ERY formulations, including nanoparticles, have emerged. Despite the interest in ERY-(nano)formulations, a review on them is lacking. Therefore, this work was aimed at reviewing all efforts made to encapsulate ERY in formulations of various chemical compositions, sizes and morphologies. In addition, their preparation/synthesis, physico-chemical properties and performances were carefully analysed. Limitations of these studies, particularly the quantification of ERY, are discussed as well.

## 1. Introduction

Erythromycin (ERY) is a therapeutical compound belonging to the class of macrolide antibiotics, originally discovered by *McGuire* et al. in 1952 [1] and later produced through biosynthesis during fermentation from species of the Gram-positive *Saccharopolyspora erythraea*, formerly classified as *Streptomyces erythraeus* [2,3]. The erythromycin group generated by bacterial strains encompasses numerous structural variations, that is, ERY A, B, C, D, E, F and G (Figure 1).

Out of the variety of ERY compounds, ERY A (ilotycin) is the primary constituent while ERY E and F are its metabolites [4]. The other forms are intermediates generated during the biosynthesis of ERY A [5].

ERY is a wide spectrum antibiotic, possessing significant therapeutic value both internally and externally. In the first case, it is administered with regard to the treatment of infectious diseases caused by mostly Gram-positive, but also Gram-negative [6] pathogens, such as *Corynebacterium diphteriae*, *Staphylococcus aureus* (*S. aureus*), *Streptococcus* spp., *Bordetella pertussis*, *Chlamydia trachomatis*, *Mycoplasma pneumoniae*, *Legionella pneumophila*, *Neisseria*, *Campylobacter fetus jejuni*, and *Bartonella henselae* [7,8,9,10,11,12,13]. When administrated externally, it is used for the treatment of acne vulgaris caused by *Propionibacterium acnes (P. acnes)*, being the most widely used antibiotic for this skin disease, due to its ability to inhibit the expression of lipase in *P. acnes* at concentrations below minimum inhibitory concentration (MIC) values [14].

Furthermore, ERY has recently proven itself useful for other types of treatment, due to its anti-inflammatory effects and ability to inhibit the formation of osteoclasts [15], being commonly used in bone cement for preventing infections [16]. Considering its role as a motilin receptor agonist, it is also used to treat patients necessitating the acceleration of gastric emptying [17,18]. It has a wide body distribution, with preponderant accumulation in the lungs; therefore, many lung-targeted formulations are being studied, especially since ERY has proven to be effective for the treatment of chronic obstructive pulmonary disease [19].

However, ERY therapy has a series of shortcomings, such as poor aqueous solubility due to its hydrophobic nature [20], instability in acidic medium (the majority of the dose is degraded at the gastric level), its relatively short half-life of about 1–1.5 h, and low bioavailability. It has been documented that ERY produces substantial gastrointestinal side effects, as well as liver toxicity, due to its instability and chemical conversion under acidic conditions [21]. Elderly patients are often unable to tolerate the gut-motility activity of the drug, whereas children find the taste unacceptable [22]. Another crucial drawback is related to its widespread distribution, which causes drug accumulation at other sites than target ones, provoking a series of undesirable adverse effects [23]. The latter-mentioned limitation was one of the driving forces in recent development of novel targeted ERY formulations.

Considering that the development of new antibiotics has reached a plateau, another issue which makes the design of improved ERY formulations a current priority is the rapidly increase antibiotic resistance. Therefore, lowering the effective therapeutic dosages by amplifying the antibiotic’s potency has become a necessity.

To address these issues, adjustments of the formulation and delivery of ERY have been previously pursued in the pharmaceutical industry. One of the most successful solutions is the structural alteration of ERY, with the synthesis of prodrugs as salts and esters, aiming for the reduction of gastric degradation and increase in hydrophilicity. The second major pathway of avoiding gastric degradation of the drug is the design of enteric coated oral formulations. Despite these efforts, issues still arise due to the fact that these attempts have not yet managed to completely improve ERY administration and decrease the side effects.

Thereby, over the past decades, a plethora of ERY formulations were developed, targeting to overcome the ERY drawbacks and to improve its bioavailability (Figure 1). Nevertheless, despite the interest for ERY-(nano)formulations, a comprehensive review on the topic is lacking. In this light, the aim of this paper was to bring together the data reported on ERY formulations, with a particular emphasis on encapsulation of the antibiotic in smart (nano)carriers, and their more complex formulas obtained by integration into gels, fibres and membranes. Their preparation/synthesis, physico-chemical properties and performances were carefully analysed, while also discussing the limitations of these studies, particularly the quantification of ERY.

## 2. Vesicles

The literature reveals a large variety of vesicular systems used to deliver drugs, such as lipid particles, liposomes, niosomes, sphingosomes, pharmacosomes, transferosomes, polymeric micelles, etc. [24]. Generally speaking, vesicular carriers are formulated with lipid materials for the purpose of overcoming challenges of drug delivery and improving therapeutic outcomes. Encapsulation of both hydrophilic and lipophilic drugs, enhancement of bioavailability, increasing the drug’s circulation time, resolving toxicity issues and lowering therapeutic dosages are some of the major benefits of lipid-based drug delivery. In the following, ERY vesicle formulations developed over a long time; their preparation and performances will be presented.

### 2.1. Liposomes

Liposomes are sphere-shaped vesicles of small diameter, ranging from a few nanometres to several micrometres, consisting of aqueous compartments surrounded by one or more lipid bilayers (Figure 2) [25].

The lipid layer is usually formed by a natural or synthetic phospholipid, but other lipids, such as cholesterol, can be incorporated in order to improve a specific property (e.g., the stability in media of different pH values of biological fluids), but also for reducing the fluidity of the bilayer membrane [26,27,28].

Reported first by the Bangham group in 1965 [29,30], the liposomes were developed as useful vehicles for protection, delivery and specific targeting of various drugs. They can encapsulate either hydrophilic medicinal substances in the aqueous compartments or hydrophobic ones in the lipid bilayers, increasing their bioavailability and circulation time in the bloodstream, therefore preventing their premature disruption by blood components. Furthermore, the lipid layer confers biocompatibility and biodegradability to the encapsulated compounds. Their potential as drug delivery systems has been demonstrated through their successful clinical applications and many preclinical trials [31,32]. Nevertheless, the investigation of ERY liposomes appears to still be in its infancy.

Liposomal ERY was first reported in 1991, within an exhaustive study which focused on the influence of the phospholipid type over the encapsulation efficiency (EE). Using the film dehydration/rehydration vesicle (DRV) method for preparation (consisting in the rehydration of the lipid/ERY film in phosphate-buffered saline (PBS) and sepharose CL-4B column chromatography for purification), the authors obtained liposomes with diameters between 250–350 nm [33]. They established that phosphatidylglycerol (PG) has a superior ability to retain ERY compared to phosphatidylcholine (PC) or phosphatidylethanolamine (PE). This was attributed to the least bulky head group of PG, therefore creating smaller steric barriers for ERY diffusion into the hydrophobic layers. When the phospholipid was composed of mixed fatty acids (saturated and unsaturated ones), the ability to retain ERY significantly increased. This was justified by the capacity of unsaturated fatty acids to improve the fluidity of the hydrophobic part of the phospholipid, making it more favourable for encapsulation of hydrophobic ERY (Figure 3). It was also discovered that the number of hydroxyl groups in the polar head of the phospholipid did not play a critical role in ERY encapsulation.

The Omri group proposed a modified protocol of the DRV method for ERY liposomal preparation, named “active loading technique”, consisting of loading the drug into blank vesicles, which were previously prepared via dehydration–rehydration (Figure 2) [34]. Using a fatty acid-based phospholipid (1,2-dipalmitoyl-sn-glycero-3phosphocholine) combined with cholesterol (2/1, *w*/*w*), they prepared blank liposomes by rehydration of the thin lipid film in water/sucrose (1/1, *w*/*w*, sucrose to lipid) followed by sonication, in order to obtain small unilamelar vesicles. Subsequently, liposomes with ERY were prepared by loading the blank liposomes with the antibiotic, washing with PBS and lyophilization. Selection of the phospholipid was justified not only by its higher ability to retain the antibiotic, but also by its ability to improve the liposome’s stability in PBS compared to other phospholipids, such as 1,2-dimyristoyl-sn-glycero-3-phosphocholine or distearoylphosphatidylcholine [35]. Sucrose was used in order to prevent liposomes deterioration during lyophilization, choosing an optimal concentration to avoid reduction of the encapsulation rate. It appears that this protocol significantly improved the encapsulation efficiency to 32%, for liposomes with a diameter of 194 nm and excellent stability. ERY remained entrapped 100% at 4 °C in PBS, 80% at 37 °C in PBS and 50% in plasma. The authors remarked that sonication of the blank liposomes favoured the yielding of smaller loaded liposomes. They attributed this achievement to the (i) use of sucrose which prevented vesicle collision and to the (ii) lyophilization process which apparently eliminates large vesicles. It was assumed that sucrose interacts with the polar heads of the phospholipids, being a better liposome stabilizer than monosaccharides and polysaccharides. Moreover, there is proof that sucrose favoured a higher entrapment efficiency, especially when it was added during both lipid film hydration and hydration of the lyophilized liposomes [36].

Nevertheless, decreasing the amount of cholesterol (1,2-dipalmitoyl-sn-glycero-3phosphocholine/cholesterol, molar ratio 6/1) and using the DRV method, the same group reported the preparation of liposomal ERY with a significant improvement in EE (~51%), but with a dramatical increase in liposomes size (~3 μm) [37]. The increase in EE values was attributed to the direct dissolution of the lipophilic ERY in the organic lipid solution combined with a prolonged sonication time. No explanation for attaining such a high size for liposomes was given. However, one reason for this could be pinned down to the interactions between cholesterol and the lipid bilayer, inducing condensation effects [38]. Compared to free ERY, the liposomal formulation showed a significant enhancement in the inhibitory and bactericidal activities against two *Pseudomonas aeruginosa* (*P. aeruginosa*) strains, with a MIC of four and eight-fold higher, which further improved when a broad-spectrum efflux pump inhibitor (phenylalanine arginine beta-naphthylamide—PABN) was added. Remarkably, the liposomal formulations showed a notable increase in the eradication of biofilms as well, indicating potential for the management of *P. aeruginosa* infections in cystic fibrosis patients.

Recent investigations revealed the multi-inlet vortex mixer technology as a suitable method to reach liposomal ERY with sizes lower than 200 nm and an encapsulation efficiency around 35%. It was reported that this method allowed for a good control on liposome size and EE by simply tuning the aqueous/organic flow rate ratio and flow parameters [39].

**Scheme 2 pharmaceutics-14-02180-sch002:**
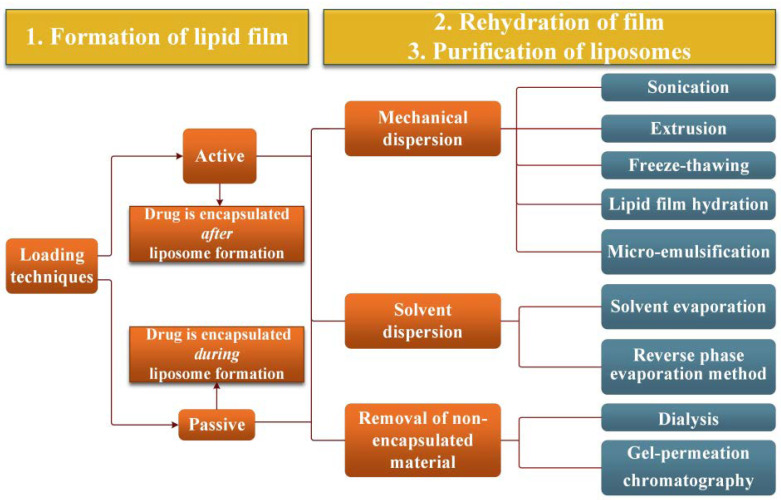
Preparation steps of liposomal ERY comprising different techniques reported in the literature for their optimization. Adaptation from [40].

Controlling the permeability of liposome membranes is an important aspect for modulating drug release in a precise manner. Starting from the premise that ***nonionic liposomes*** facilitate the penetration of drugs through the skin barrier more efficiently than phospholipidic liposomes, liposomal ERY was prepared using a mixture of non-ionic lipids: cholesterol (CH), glyceryl dilaurate (GDL), and polyoxyethylene-10-stearyl ether (POE) [41]. The preparation method involved the dissolution of ERY in a mixture of melted lipids (GDL:CH:POE, 57:15:28. *w*/*w*/*w*) followed by rehydration in isotonic HEPES buffer. Unfortunately, no information about the size, stability and EE of the resulting liposomes was provided, and the effectiveness of this preparation method could not be evaluated. They were more effective for ERY uptake into living skin compared to those based on phospholipids, due to the ability of GDL and POE to function as permeation enhancers, suggesting the nonionic liposomes as more suitable for topical formulations.

With the purpose of being utilized in the sphere of pisciculture, liposomal ERY was prepared via a slightly modified DRV method, using a combination of lipids (phosphatidyl choline/cholesterol/phosphatidyl glycerol, 10/4/1, *w*/*w*/*w*) [42]. The liposomes had micrometric size and high polydispersity degree, their diameters covering values in the range of 1–40 μm, with a prevalence of 2–3 μm sized liposomes. They were later bioencapsulated into brine shrimp *Artemia franciscana* with the intent of targeting Pacific salmon kidney disease of bacterial origin, creating a more palatable live feed. Even though no direct information was provided regarding the EE, the low ERY concentration in artemia indicated a low drug encapsulation within the liposomes.

A recent paper reported the preparation of liposomal ERY by an active loading technique method, followed by coating with chitosan [43]. The resulted liposomes showed a hydrodynamic diameter of 311 nm and an EE of 55%. It appeared that a dramatic increase in Zeta potential was induced by chitosan coating (37 vs. 0.16 mV), suggesting the stabilization of the liposomes. The investigation of their biocompatibility, performed on mice, showed no toxic effects subsequent to oral administration.

The modern trend in drug delivery is directed towards the development of ***smart targeted drug release in response to various stimuli***, such as pH, temperature, light, ultrasound, magnetic field and specific receptors from disease sites [44]. This can ensure the liposomes delivery to the targeted areas of disease, avoiding large drug dosages [45].

Being intended for the treatment of acne, ERY-loaded liposomes decorated with a photosensitizer were designed in order to ensure a chemically controlled release under the influence of *P. acnes* lipases, as an environmental stimulus. Therefore, a pheophorbide photosensitizer was linked with pullulan via an ester linkage (Pu-Pheo) while 1,2 dipalmitoyl-sn-phosphoatidylcholine was chosen as a constituting lipid, knowing that lipase specifically breaks their ester bonds [46]. The liposomes were prepared by DRV method, using erythromycin/1,2 dipalmitoyl-sn-phosphoatidylcholine/cholesterol (1/10/1.5, *w*/*w*/*w*). Preparation steps included rehydration in ammonium sulfate followed by decoration with PU-Pheo, and finally by lyophilization using sucrose lipoprotectant. The PU-Pheo-decorated liposomes had a quite small average hydrodynamic diameter of 93.6±3.1 nm. The in vitro release tests, carried out by dialysis in PBST (0.05 Tween 20 in PBS), showed a sustained release, achieving a 90% ERY cumulative release in 48 h, compared to free ERY which was released entirely within 12 h. The transdermal release evaluated via a Franz cell system using mouse skin demonstrated that ERY was preferentially delivered into infected inflammatory skin dermis, confirming the selective disruption of the liposomes in *P. acnes*-rich areas. Even more, the PU-Pheo-decorated liposomes proved a higher antibacterial effect in vitro under laser irradiation, indicating an additional antibacterial effect induced by the photosensitizer. These results were confirmed by in vivo tests on mice, which revealed that the combined treatment with both liposomes and laser irradiation decreased the number of residual *P. acnes* bacteria to an undetectable level, whereas each individual therapy was less effective. Importantly, no toxic effects were detected. These findings suggested that the combined therapy is a valuable route to achieve a reduction of the minimum therapeutic dose, which is promising for preventing the phenomenon of antibiotic resistance.

The same authors reported other similar studies focused on designing liposomes bearing a polycationic photosensitizer, polyethyleneimine-hematoporphyrin (PEI-HP), which was chosen due to its potential to interact with negatively charged sites at the bacterial surface [47]. The liposomes were prepared by an analogous procedure, with slight modifications, aside from the fact that the photosensitizer was mixed with the lipids and ERY in order to form the lipidic film. It was noted that this procedure led to liposomes with a higher average diameter, ~200 nm, a loading efficiency of ~85% and a loading content of ~7%. It was assumed that the high dimension of PEI determined the increase in liposome size, while its polycationic character improved the stability (Zeta potential approximately +52 mV). Following similar in vitro and in vivo drug release investigations, the authors proved that the ERY liposomes containing PEI-HP were effective in suppressing *P. acnes* by a dual phototherapeutic mechanism.

Another stimulus which was examined for its potential to achieve a targeted release of ERY from liposomes was the ***magnetic field***, considering that the mechanism of magnetic triggering is one of the most potent strategies for designing targeted release formulations [48]. Hence, magneto-liposomes containing ERY were proposed as a combined strategy for controlled drug release, based on the premise that the electromagnetic field improves the permeability of the phospholipid membrane [49]. ***Magneto liposomes*** were prepared using iron oxide nanoparticles which are FDA approved for medical applications [50,51]. As for their preparation, the selected method was that of solvent injection, which involved dissolving the soybean phosphatidyl choline, ceteareth-25 edge activator, and ERY in ethanol at 30 °C, followed by the slow addition of an aqueous suspension of magnetic nanoparticles (Fe_3_O_4_) under continuous magnetic stirring and sonication [49]. The resulted magneto-liposomes had a mean hydrodynamic diameter of 473±10 nm, which was significantly higher compared to that of the corresponding unmodified liposomal ERY (254 ± 10 nm). However, no difference was noted for the Zeta potential values, which were ~47 mV in both cases, indicating no influence of the magnetic nanoparticles over the liposome’s stability. Moreover, the encapsulation efficiency appeared to be favoured by the magnetic nanoparticles, from 87.9 ± 2.2% for the corresponding unmodified liposomal ERY to 97.8 ± 1.5% for magneto-liposomal ERY. In vitro drug release experiments demonstrated that the ERY release was induced at a low-intensity magnetic field, and the release rate was controlled by varying its intensity. The authors hypothesized that the integration of iron-based nanoparticles into the lipid layer produced an overall stiffening and decrease in homogeneity. The application of a magnetic field caused rotations and vibrations of the iron nanoparticles, generating mechanical forces, which induced deformations of the phospholipid membrane, altering its integrity.

Amongst the macrolide liposomes, ERY ones appear to have the best chances for application. This was indicated by some computational investigation of the impact of macrolide liposomes on living organisms, which suggested that phospholipids can interact with cellular components, inhibiting enzyme activity and consequently leading to the lipid accumulation in cells, raising potential concerns for toxicity. Among the studied macrolides, ERY showed the lowest negative impact [52].

### 2.2. Ethosomes

To improve drug delivery performance of liposomes, soft vesicles, mainly containing molecules with the capacity to induce more fluidity into the phospholipid bilayers, were proposed. When ethanol (20–50%) is entrapped into liposomes, the soft vesicles are called ethosomes [53]. Compared to liposomes, which remain confined to the upper *stratum corneum* of the skin, ethosomes exhibit a higher potential to transport active therapeutic agents into deeper skin layers, due to the permeation enhancing capacity of ethanol [54,55]. It was stated that ethanol can fluidize both the vesicles and the *stratum corneum* lipids, making the skin more penetrable, while the vesicles become more flexible.

ERY ethosomes were prepared in a well-sealed container by dissolving phospholipon 90 and ERY in ethanol, followed by the slow addition of distilled water under constant stirring at 30 °C [56]. They displayed a spherical unilamelar configuration with low size (116 and 123 nm), EE of ~78%, improved bilayer fluidity and excellent shelf-life, their size and configuration being almost unaffected after one year of storage under ambiental conditions (Figure 4). The investigation of the antibacterial properties against *S. aureus* and *B. subtills* showed improved activity when compared to standard hydroethanolic solutions, with remarkable reduction of the MIC of 2.5 times, against a clinically isolated *S. aureus* strain with known resistance to ERY. These results indicated an improvement of ERY delivery from the ethosomes to the bacteria, which was attributed to an enhanced permeability of microorganisms to ethosomal ERY, suggesting the necessity of administrating lower doses of the drug. Moreover, this study suggested that ethosomes are a good approach to overcome bacterial resistance to antibiotics.

The ethosomes were not toxic to dermal fibroblasts and did not show any skin irritation. In vivo experiments on mice intradermally inoculated with live *S. aureus* revealed that the topical administration of ethosomal formulation (ethosomal ERY dispersed into a Carbopol gel) was as effective as systemically administrated ERY, resulting in efficient infection inhibition and healing *S. aureus*-induced deep dermal [57]. Moreover, the histological investigations revealed no bacterial growth and a normal skin structure, whilst topical application of a hydroethanolic ERY revealed no subdermal healing and necrosis destroying the skin structure. All data indicated that the ethosomes are efficient carriers for ERY delivery into the deep skin strata for the eradication of staphylococcal infections.

### 2.3. Niosomes

Niosomes are slightly modified version of liposomes, in which the lipid is replaced by non-ionic surfactants (Figure 5) [58]. They are biodegradable, relatively nontoxic, and more stable, yet less expensive compared to liposomes. Drug-loaded niosomes show interactions with the epidermal tissue without presenting an immediate or strong systemic action, thus being preferred for dermal applications.

An exhaustive study was presented on the optimization of niosomal ERY preparation by the thin film hydration technique, focusing on the surfactant type, cholesterol/surfactant ratio, volume and time of hydration, stirring speed, vacuum intensity, temperature, annealing time, solvent system and solvent volume [59].

The optimized niosomes (ERY/Cholesterol/Span 80 molar ratio of 1/1/2) reached a drug entrapment of ~90% and the smallest size of 4.5 µm. Their suspension presented good stability at 4 °C, with a drug leakage of 35% after 12 weeks, but low stability at room temperature with a drug leakage of 83% over a similar period. This was justified by the higher fluidity of the lipid bilayer at higher temperatures.

Varying the previously mentioned parameters in a broader range, the same authors obtained niosomes with similar drug entrapment (88%) for a different composition (ERY/Span80/cholesterol molar ratio of 0.625/1.5/1), highlighting the high impact of preparation conditions [60].

Applying a slightly different procedure, another group prepared niosomal ERY (Span60/Cholesterol in a mass ratio 70/30) and used them in clinical trials for the treatment of mild to moderate acne vulgaris [61]. The trial showed the superior performance of the niosomal ERY (4%) in comparison to a control (ERY (4%) and zinc acetate (1.2%)), with more than 75% reduction in the number of lesions observed for 26.6% patients (Figure 6) and much less severe side effects than in the control group.

### 2.4. Micelles

Micelles are obtained by similar preparation methods and composition as niosomes, with the difference that cholesterol is absent (Figure 7). Liu et al. developed a micelle formulation encapsulating ERY with the use of ***Pluronic F-127*** nonionic surfactant [62]. The obtained micelles had an EE of 28.3%, and they displayed a spherical morphology with an approximate diameter of 193 nm. In vitro release studies concluded that over 90% of the drug was released over 8 h, but the antibacterial activity tested on strains of *S. aureus* and *Escherichia coli* (*E. coli*) revealed MIC values five times higher than those of simple ERY.

Another group aimed to prepare ***Pluronic*** micelles incorporating ERY, generally focusing on the size distribution and aggregation of the particles in dilute aqueous media. Compared to the previous study, the particle size was considerably smaller (70 nm). The study demonstrated a strong pH dependence of the micelle behaviour. A pH increase from 4.65 to 6.1 led to the ionization of ERY and consequently to the formation of larger sized micelles, while an increase in pH to 11.36 triggered the micelle rupture and consequently the drug release [63].

Using a block copolymer with amphiphilic properties (*poly(ethylene glycol)-block-poly[2-(2-methoxyethoxy) ethyl methacrylate*)], polymeric micelles encapsulating ERY were prepared via dialysis method. They had a loading degree (LD) of 16.8%, size of 741.2 nm, negative Zeta potential, and were able to release the drug over the timespan of 180 h [64]. Using other amphiphilic copolymer, PCL-PEG, ERY micelles with LD of 6.5%, EE of 97%, size of 220 nm, and positive Zeta potential (+19 mV) were prepared. The formulation was proven to lack cellular toxicity and to successfully inhibit the proliferation of *S. aureus* species through a sustained release [65].

### 2.5. Cubosomes

Cubosomes are discrete, sub-micron nanostructured particles comprising a bicontinuous cubic liquid crystalline phase, possessing a good ability to penetrate the skin even through the pilosebaceous units [66]. They are obtained by the self-assembly of certain surfactants and lipids in water, enabling the entrapment of hydrophobic and hydrophilic drugs due to their bicontinuous nature (Figure 8).

Cubosomes form a viscous optically clear material, suitable for cosmetic applications, and generally for transdermal drug delivery. In this context, the entrapment of ERY into cubosomes was regarded as an innovative alternative to the other lipid-based formulations aiming to minimize side effects while preserving its efficacy in acne treatment. Cubosomal ERY was prepared employing the emulsification method, by adding hot water, under stirring, to a mixture of glyceryl monooleate, poloxamer surfactant and ERY [67]. The optimization of glyceryl monooleate and poloxamer concentrations, stirring speed and time led to ERY-loaded cubosomes with EE values of 95% and an average particle size of 264 nm. The particles were observed to have a spherical shape along with a smooth surface and forming a gel with pseudoplastic behaviour. ERY cumulative release was 75% in the first 7 h and reached 88% after 24 h (pH = 6.4).

### 2.6. Solid Lipid Nano(Micro)Particles

Literature data mentioned the preparation of solid lipid nanoparticles (SLN) and solid lipid microparticles (SLM), the classification being created by their dimension. Nevertheless, the similarity of the composition of SLNs and SLMs induce similar physicochemical properties, the main distinction being their application fields and administration routes that are dictated by their different size [68]. Furthermore, the use of the term “nano” has varied along the course of time [69,70]. Therefore, in this section, solid lipid nano(micro)particles shall be presented as they were reported, rather than their differentiation based strictly on size criteria.

The distinct characteristic of the solid lipid nano(micro)particles is the presence of biodegradable lipid-based components found in a solid state at physiological temperatures (Figure 9). Some examples of such components are hard fats such as glycerol (mono-, di-, and/or triglycerides), complex glyceride mixtures, fatty acids, steroids or even waxes [71]. One of the obvious advantages of solid lipid particles is that the lipid matrix is composed of physiological lipids, thus significantly decreasing the risk of toxicity [72]. Therefore, the mentioned drug delivery system was promoted as a safer option, in addition to the higher stability of the solid lipid matrix compared to other nano(micro)particles, and the cheaper production costs than those of phospholipid-based liposomes developed thus far [73]. They offer great potential for the administration of active molecules by any route, including tablets and suppositories.

SLNs and SLMs consist of three main components: solid lipids, surfactants and water. Due to the solid nature of the lipid, the drug usually requires to be incorporated within the lipid after it is brought to a melted state. In this light, an important aspect that should be considered in the lipid selection is ERYs solubility in the solid lipid. A study of the partitioning behaviour of ERY in different lipids showed that it presents the best miscibility when added along with glycerol monostearate, followed by stearic acid, acetyl alcohol, and cetostearyl alcohol [74]. Nonetheless, other solid lipids and mixtures of solid lipids were used with good results.

#### 2.6.1. Solid Lipid Nanoparticles

An exhaustive study regarding the preparation and composition optimization of ERY loaded solid lipid nanoparticles (ERY-SLN) was realized by using the hot homogenization method choosing *glycerol monostearate* to act as the lipid phase, Poloxamer 188 as surfactant and soya lecithin as a cosurfactant [74]. Even though no other component was mentioned, the DSC study makes reference to the presence of mannitol as a cryoprotectant.

Considering that the surfactant concentration can shape the properties of SLNs, different compositions were investigated, and it was found that 2% (*w*/*v*) Poloxamer 188 is optimal to reach the smallest particle size of 138 nm, with the highest EE and DL (84.5% and 28.1%, respectively). Studying the effect of different variables on SLN properties via a miscellaneous design–response surface methodology, the authors concluded that the increase in lipid concentration produced a decrease in the mean particle size but also in the entrapment efficiency. Therefore, a minimum particle size of 153 nm and maximum EE of 88.4% were reached when using a lipid concentration of 15 mg/mL and surfactant/cosurfactant ratio of 1.07/1. The ERY-SLN presented negative Zeta potential values up to −19 mV, reflecting an excellent stability. No significant alteration of the particle size and EE was observed for the SLN dispersions stored at 4 or 25 °C, over a three-month period. The release profile, which was investigated in phosphate buffer media, showed a prolonged release during 24 h, up to 66.26 ± 2.83%.

Pursuing the preparation of a topical formulation aimed at the treatment of acne, *Dhillon* et al. investigated the development and optimization of ERY-loaded SLNs. The study focused on selecting the best preparation method, lipid type, drug/lipid ratio, stirring, and sonication time [75]. The findings led to the conclusion that the microencapsulation technique was superior to the solvent evaporation and the solvent emulsification diffusion ones, because it showed better results in terms of particle size, particle shape and drug entrapment. The screening of three lipids, stearic acid, glycerol monostearate and Compatriol 888, revealed that the use of *glycerol monostearate* significantly decreased the particle size into the nanometric range and favoured the highest ERY entrapment. Furthermore, it was discovered that an ERY/glycerol monostearate ratio of ½ was optimal for a maximum encapsulation of 78% into nanoparticles of 176 nm. The authors also reported that a 0.5% concentration of surfactant (Poloxamer 188) was enough to preserve the size of the nanoparticles and to ensure a high Zeta potential, required for particle stability. An increase in stirring time (from 30 to 60 min) produced a dramatic decrease in particle size (from 528 to 176 nm) and higher entrapment degree values, which reached 78.59%. A sonication time of 20 min was optimal to reduce the particle size. The average particle size, PDI, Zeta-potential, EE and LD of the optimized SLNs were found to be ~176 nm, 0.275, −34 mV, 73.56% and 69.74% respectively. They showed a prolonged release, reaching around 85% cumulative ERY release after 12 h and 93% after 24 h.

Targeting the improvement of ERY’s penetration into bacterial cells, cationic SLNs were designed using a cationic lipid, quaternary ammonium salt and didecyldimethylammonium bromide [76]. SLNs with mean diameter from 250 to 400 nm were prepared using the solvent injection method. The cationic lipid induced positive Zeta potential and great stability over time, with no change in the morphologic parameters being recorded after one year of storage. As proof of concept, a higher antimicrobial activity of SLNs compared to free ERY was evidenced, and the MIC decreased along with the increase in cationic lipid content, proving its synergic antibacterial activity.

#### 2.6.2. Solid Lipid Microparticles

ERY-loaded SLMs were prepared for oral delivery, using a binary mixture of ½ ratio solid lipids (Softisan154 and Phospholipon90H or Softisan154 and beeswax) forming a solidified reverse micellar solution (SRMS) matrix. PVA, sorbitol and sorbic acid were also used as surfactants [77]. Priorly, the SRMS matrix was prepared by fusion, and the SLMs were prepared by hot homogenization method, varying the ERY concentration in order to obtain SLMs with different drug contents.

The DSC investigation suggested the occurrence of interactions between the two lipids, leading to a less crystalline morphology, envisioned to have a higher ability to retain ERY over time compared to a more crystalline matrix, which is prone to a faster drug expulsion. The SLMs showed a spherical morphology, with a smooth and non-porous surface, while the diameters increased proportionally from 13 to 17 µm along with the entrapped ERY amount, with no significant distinctions between those containing different lipid mixtures. The nature of the lipids influenced the EE and LD, i.e., the SLMs composed of Phospholipon90H displayed higher EE values, ranging from 89 to 95%, while those containing beeswax showed variations of the EE from 78 to 89%. They were stable over the span of 4 weeks; however, an aggregation tendency towards the end was observed, while a slight pH shift towards acidity suggested the degradation of the lipid component to fatty acids.

The in vitro release studies revealed that all ERY-loaded SLMs possessed higher prolonged release properties and no burst effect or dose dumping, compared to the reference ERY-tablets. This suggests a reduction in the daily number of administrations (from 3–4 to 1–2) needed to reach and maintain the optimal plasma concentration. The release profile was dose dependent and was influenced by the nature of the lipid; SLMs containing Phospholipon90H exhibited a maximum release ranging from 90 to 75%, while those containing bee waxes ranged from 79 to 61%. The antimicrobial activity tests against *S. aureus*, *S. paratyphii*, *P. aeruginosa*, and *E. coli* confirmed a time- and dose-dependent anti-bacterial activity, higher for those containing Phospholipon90H compared to bee waxes and simple ERY. The findings suggest that the formulation of ERY as SLMs potentiates its activity, possibly due to the fusion between the lipid component and the bacterial cell membranes, promoting an easier penetration of the antibiotic. Pharmacokinetical studies on rats showed a steady and slow decrease in the drug concentrations or a gradual clearance, reaching a T_1/2_ value for the SLMs of 8 h, compared to a T_1/2_ of 4.5 h in the case of the neat ERY.

ERY-loaded SLMs were also prepared using *stearic acid* as the lipid component and Tween 80 as surfactant, via hot homogenization method [78]. By varying the lipid/surfactant ratio, the average hydrodynamic diameters ranged from 2.28 to 8.15 μm, the lowest particle size being reached for a lipid/surfactant ratio of 1 g/1 mL and 1 g/2 mL. The melting temperature of ERY was clearly identified in the DSC curves, indicating that it was dispersed as large particles into the lipid, attributed to the ERY insolubility into the lipid matrix. The microparticles showed much higher in vitro antimicrobial activity compared to neat ERY; however, those with the lowest MIC values were large in size (6.5 μm), at risk of not being able to pass into the lymphatic circulation [79]. Furthermore, the microparticles containing higher amounts of surfactant and thus more stable in acidic environment, were selected for in vivo tests on model infected animals. It was proven that the administration of equal amounts of SLM and neat ERY led to almost similar bacteremia reduction (75% vs. 72% during day 3, and 95% vs. 94% during day 7), despite the small percentage of ERY entrapped into SLMs (21%). This suggested a significant increase in the antimicrobial activity of ERY when encapsulated into SLMs. Further, only low fluctuations in plasma concentrations were recorded in the case of SLMs, indicating a potential to reduce major side effects.

## 3. Polymer Micro(Nano)Spheres

Microencapsulation is a term used to describe the process of incorporating tiny particles into a polymeric matrix, in order to protect specific functional materials and release them in a prolonged manner (Figure 10). The polymers, of synthetic or natural origin, play the role of a functional matrix, controlling the diffusion and release of the encapsulated compound followed by their elimination from the system [80,81]. Seeing as one of the basic requirements for any pharmaceutical ingredient is biocompatibility along with an easy removal from the body, either through metabolization or kidney filtration, polymers which undergo degradation in a living environment are therefore preferred.

### 3.1. Microencapsulation in Synthetic Polymers

Due to the lipophilic nature of ERY, the most suitable method for its encapsulation involves emulsion solvent evaporation performed in an aqueous environment [80]. There have been many attempts to achieve ERY microcapsules targeted for specific applications, mostly by using various polymers as coating matrixes. Some of the most relevant are described below, as follows.

Designing ERY microparticles for in vivo applications, biodegradable ***poly(1-lactide)* (PLA)/*poly(ethylene glycol)* (PEG)** copolymer, containing 5% and 10% PEG, respectively, was proposed as a matrix constituent [84]. The choice was justified by the high solubility of PEG, which has the ability to improve the hydrophilicity of PLA and thus the diffusivity of water and drug molecules through the polymer carrier [85]. The morphological determinations showed that the PLA/PEG microcapsules had a larger size compared to PLA ones (57 µm vs. 40 µm) and a rougher surface, attributed to the higher swelling ability induced by PEG. However, the presence of PEG improved the loading efficiency, from an initial 42% in the case of PLA to 48 and 56% for PLA/PEG containing 5% and 10% PEG, respectively. Furthermore, the inclusion of PEG improved the biodegradation rate, reaching a mass loss of ~60% in three weeks, for the PLA/PEG microparticles containing 10% PEG, compared to the PLA homopolymer which showed a much smaller rate of ~30% in a similar period. This alteration affected the drug release rate, reaching 2.7 ppm for PLA/PEG 10% microparticles versus 1.4 ppm for the PLA ones, after 30 h.

The same group also reported ERY microcapsules prepared with ***poly(******ɛ******-caprolactone)***
**(PCL)/*poly(ethylene glycol)*(PEG)**, following a similar procedure for the manufacturing and characterization of the formula, as previously described [86,87,88]. In this case, PVA, Span 80 and gelatine were separately used as emulsifiers, while the PEG content varied between 0–50%. A comprehensive investigation of the microcapsule morphology after varying several preparation conditions established that the gelatine emulsifier favoured the formation of the smallest microcapsules (~50 μm compared to ~65 μm in the case of PVA and ~75 μm in the case of Span 80, when using 5% emulsifier). Moreover, when the emulsifier concentration (gelatine) was higher than 2% (3–6%), spherical, stable microcapsules were generated, coupled with a proportional decrease in their diameter, as the emulsifier concentration increased up to a certain percentage. For instance, it was stated that once the gelatine concentration was increased from 1 to 5%, the average size decreased from 68 to 6 μm, while a concentration of 6% determined the diameter to increase to 14 μm. This effect was attributed to the aggregation of the emulsifier around the microcapsules at higher concentrations. The authors also noted that the microcapsule’s diameter could be controlled by adjusting the stirring velocity, i.e., the diameter diminished as the stirring intensified. In this regard, it was concluded that a stirring rate of 11,000 rpm led to a reduction of the microcapsules diameter to 0.2 μm. As a result of increasing the PEG content into the PCL/PEG blend, the hydrophilicity and hence biodegradability improved, favouring a faster ERY release in buffer solution, a significantly higher cumulative release being recorded over 25 days for the PCL/PEG matrix compared to the neat PCL one.

A recent article reported ERY microcapsules based on ***poly(DL-lactide-co-glycolide)*** (PLGA) as the polymeric matrix [89]. The preparation method used propylene carbonate (PC), which is a polar aprotic solvent, followed by the addition of polysorbate acting as emulsifier. The obtained ERY microparticles were large sized (59 μm) and presented 38% encapsulation efficiency. The in vitro release studies showed that ERY was delivered in a prolonged manner over 20 days, with an initial burst effect of approximately 50% within the first 2 h. It was stated that the high polarity of PC favoured the dissolution of PLGA and ERY and thus created the premises for good drug dispersion. A notable finding of this study is that ***the microparticle morphology can be manipulated by means of a proper preparation temperature***, which depends mostly on the T_g_ values of the polymer/drug. Another useful observation was made, related to the connection between the drug’s nature and the EE, remarking that ***a better miscibility between matrix and drug fosters an increased EE***.

Remarkably, a combination of ***PLGA/PCL mixture*** was used to prepare ERY microspheres by the emulsion solvent evaporation method, starting from the premises that these two polymers form an amphiphilic system (PCL—hydrophilic; PLGA—hydrophobic) [90]. The ERY release profile revealed a rapid delivery from PLGA/PCA-based microspheres compared to PCL ones by virtue of the increased hydrophilicity. The greater ERY release was also confirmed by the improved antimicrobial activity against *S. aureus*.

### 3.2. Microencapsulation into Polymers of Natural Origin

A noteworthy study aimed to encapsulate ERY into a biopolymer (***gum Arabic, maltodextrin***) via the one step spray-drying technique [91], consisting in the dispersion of ERY and a lipid into a polymeric mixture which was then sprayed and dried in an atomizer. It was stated that microparticles with high EE values (80%) and diameters less than 100 µm were obtained for an optimal ERY/polymer excipient ratio of 30/70. This method appears to be of great importance for the reason that it avoids the use of potential toxic organic solvents, but also because it consists of simple operations and can be easily scaled-up.

Commencing from the idea that microspheres with sizes ranging from 7–25 μm can be concentrated in the lung through i.v. administrations, a lung-targeted formulation of ERY was prepared by encapsulation into ***gelatine*** microspheres through a double emulsion solvent evaporation process [19]. Gelatine was chosen as the matrix for ERY due to its advantageous properties, such as nontoxicity, biodegradability, biocompatibility and nonimunogenicity. Nevertheless, it is a natural inexpensive polymer which has the benefit of being FDA-approved for i.v. administration in both powdered and solution forms [92,93]. The SEM analysis revealed microspheres with a smooth, porous surface and an average diameter of 15.62 μm (Figure 11). They showed LD of 13.56 ± 0.25% and EE of 55.82 ± 2.23%. The DSC curve only exhibited the melting point of gelatine shifting to a lower value, suggesting that ERY was intimately dispersed into the gelatine matrix. In vitro release tests revealed an ERY delivery of 80% over 4 h, aided by the hydrophilic nature of gelatine. Following administration to rabbits, neither vein irritation nor haemolysis were noted, while the tissue distribution recorded after 2 h of administration indicated a selective accumulation in the lungs. ERY pulmonary concentrations were 15.92 times higher than those found in plasma, compared to the ERY control which was only 3.51 times higher. Moreover, the authors stated that in the case of microcapsule administration, ERY concentrations in the pulmonary tissue were higher than those in liver, kidney, spleen, and heart tissues, whilst similar concentrations in all compartments were recorded for the administration of ERY control. Compared to the control test, the Drug Targeting Index of gelatine microsphere formulations was 6.51. The authors further pursued the in vivo investigations, replacing healthy rabbits with *Mycoplasma pneumoniae*-infected rats. When the microparticles were administrated through the intragastric route, a significant alleviation of the symptoms was recorded, along with a dose-dependent reduction of the pulmonary index in infected rats [94].

Utilizing the nanoprecipitation method, ***gelatine*** was also used to prepare ERY nanocapsules [95]. First, gelatine nanoparticles were prepared using Pluronic surfactant and crosslinking with glutaraldehyde, and then, they were loaded with ERY by incubation into a saturated ERY solution. The resulted ERY–gelatine nanoparticles were spherical, with an average hydrodynamic diameter of 175 nm and a very high Zeta potential of -59 mV, indicating a strong anionic character and high stability. ERY was released from the nanoparticles in a prolonged manner, reaching 86% after 72 h, while the antibacterial studies displayed a high activity against strains of *S. aureus* and *P. aeruginosa*, with inhibition zones comparable to those of standard antibiotic discs of Cefoxitin and Ciprofloxacin or even larger.

In perspective of designing a formulation for the treatment of cow mastitis, ERY-loaded ***chitosan*** nanoparticles were prepared by ionotropic gelation [96]. The chitosan matrix was selected considering its proven intrinsic activity, both in vitro and in vivo, against *S. aureus* isolated from cows with mastitis, along with its good mucoadhesive properties [97]. The preparation involved suspending ERY in chitosan solution, emulsification with Tween 80, and crosslinking with an aqueous solution of TPP. SEM images revealed spheres with a mean diameter of ~241 nm and DLS analyses indicated a Zeta potential of ~23 mV for ERY-loaded nanoparticles compared to ~140 nm and ~29 mV for the neat chitosan ones, suggesting the occurrence of some sort of ***intermolecular forces between ERY and chitosan***. The authors claimed a dramatic increase in the entrapment efficiency after freeze-drying, from 2.45 ± 2.33 (%) for the initial suspension to 80.37 ± 1.98 (%) for the lyophilized sample resuspended in distilled water. The in vitro release tests showed a much slower and prolonged release for both suspension and powdered forms compared to free ERY, reaching a cumulative release of ~50% in the first 16 h compared to ~93% for free ERY. This finding was confirmed by the ~3 times higher antibacterial activity of the ERY nanoparticles compared to that of free ERY, revealed by tests against *S. aureus*, on both MRSA and MSSA bacterial strains. Nevertheless, the plain chitosan nanoparticles also showed remarkable antimicrobial activity, indicating a synergy between the two components.

Ionotropic gelation was also used for the preparation of ERY ***chitosan*** nanoparticles, using a slightly different method, mostly consisting of mixing ERY into a chitosan solution followed by the addition of TPP [98]. This minor alteration appeared to have a strong impact on the nanoparticles size, decreasing the average diameter to ~50 nm, but also on other characteristics such as Zeta potential (~14 mV) and EE (95%). The nanoparticles showed high antibacterial activity against *S. aureus*, *E. coli* and *P. aeruginosa*, exhibiting a stronger activity against Gram-positive strains, with MIC values significantly lower than those of neat ERY and chitosan. This strengthens the idea of the existence of a ***synergic effect between ERY and chitosan*** when combined in ERY-NP, therefore encouraging further studies.

The same ionotropic gelation method was applied for a mixture of chitosan/hydroxyapatite/ERY to prepare nanoparticles of 100 -200 nm, with LD of 16% and Zeta potential 18 [99]. The nanoparticles inhibited the growth of *Bacillus cereus* (*B. cereus*) and *Salmonella*
*entrica* and were able to remove their biofilms. No haemotoxicity was recorded, therefore being proposed as ERY delivery systems for osteomyelitis treatment.

One more naturally originating polymer that can be manipulated to produce ERY nanoparticles is ***dextran*** [100]. A pH-sensitive formulation was created, pursuing this idea through the derivatization of dextran with glycidyl methacrylate (dex-GMA) in various proportions, leading to products with different degrees of substitution. The ERY particles were prepared by three methods, all of them relying on the emulsion polymerization technique, the distinction between them consisting in the adjustment of preparation conditions (reagents, temperature, time, etc.). They provided NPs with different sizes (100 or 50 nm) and extremely low LD values, namely 1.06%, 0.7% and 0.5%, showing that the particles were poorly loaded with ERY.

An interesting study reported loading ERY into natural ***sporopollenin microcapsules***, extracted from *Lycopodium clavatum* spores (LCS) and processing them in order to obtain nitrogen-free LCS exines (Figure 12) [101].

The drug loading was carried out by immersing the LCS microcapsules into a hydroalcoholic ERY solution, in order to allow for the passive diffusion of ERY, followed by drying. SEM images showed that the LCS microcapsules preserved their native shape, with a constant diameter of ca. 25 µm, and a reticular microstructure characterized by the presence of hexagonal cells on the surface, indicating a robust resistance of the biopolymer shell. No ERY crystals were observed on the LCS surface, while the EE was 32%, and the LD was 16%. The ERY-loaded microcapsules showed significantly improved antibacterial activity against *K. pneumonia*, *P. aeruginosa* and *S. aureus* when compared to that of pure ERY. This was attributed to the unique natural structure of the LCS surface which can favour adherence to the bacterial cell wall, forcing an enhanced contact time with the antibiotic. Biocompatibility tests on human epithelial colorectal adenocarcinoma cells Caco-2 indicated no significant toxicity of both empty LCS capsules and ERY–LCS capsules, recording a cell viability of 87.5% for a concentration of 200 mg/mL. The in vitro drug release tests in PBS showed that 75% ERY was released in the first 8 h, and continued to 85% after 40 h. In vivo release studies on rats revealed the improvement of ERY’s bioavailability in plasma, with a T_1/2_ of 2.53 h. These results indicated that ***LCS capsules could be a great alternative to encapsulate ERY*** in order to improve its bioavailability and antibacterial activity, for a safe drug delivery, surpassing the harsh gastric conditions.

Somehow, a similar study encapsulated ERY into ***artificial Janus micromotors*** which appear to be a challenging pathway in the design of controlled release drug delivery systems. Essentially, under certain environmental conditions, they are able to convert chemical energy into autonomous motion to obtain targeted release [102]. Janus ERY micromotors were prepared using anionic polyelectrolyte sodium polystyrene sulfonate (PSS), cationic polyethyleneimine (PEI), platinum and silica microspheres as sacrificial material [103]. Briefly, ten bilayers of PSS and PEI were alternatively deposited on silica spheres with an aminated surface through a layer-by-layer technique, starting with PSS, and ending with PEI. Subsequently, they were asymmetrically coated with poly(dimethylsiloxane) loaded with Pt nanoparticles by the microcontact printing method (Figure 13). Further, the silica was removed using hydrochloric acid, and the resulting hollow microcapsules were loaded with ERY, reaching ~40% drug loading. The idea behind this design was that Pt can catalyse the decomposition of the hydrogen peroxide generated by bacteria under oxidative stress, generating bubbles, which enable the micromotor movement, a phenomenon called “propellant mechanism of bubble driven micromotors”. Thus, in the context of a microbiologic environment, the ERY-loaded micromotors will move, promoting their diffusion in the surroundings and delivering ERY. The in vitro release tests (pH = 6.5 and 7.4) showed a burst effect reaching 75% cumulative release in 10 h, regardless of the pH value. When encapsulated into a hydrogel, ERY micromotors showed antibacterial activity against *E. coli* and *S. aureus* with evidently higher inhibition zones compared to ERY capsules lacking Pt, highlighting the active role of the micromotors. Furthermore, the hypothesis of the active role played by the autonomous mobility of micromotors was supported by (i) in vitro motion experiments in which the Janus micromotors were able to move when introduced in H_2_O_2_ medium and furthermore by (ii) the fact that micromotors were observed at the edge of inhibition zones at the end of the antimicrobial tests.

## 4. ERY–Cyclodextrin Complexes

The development of ERY–cyclodextrin complexes has been reported to be a promising path towards resolving a series of issues which are aroused in the formulation of ERY, such as improving solubility, reducing degradation and enhancing bioavailability, while reducing the risk of toxic effects. Among cyclodextrines, β–CD is the most frequently used due to its low price and lack of toxicity, in addition to the fact that it is recognized by GRAS as being safe for use in the food industry [104].

The *Song* group prepared cyclodextrin–erythromycin (CD–EM) complexes via two methods, the solvent evaporation and the kneading technique, aiming to obtain ERY formulations suitable for covering prosthetic surfaces in order to prevent orthopaedic infections and aseptic loosening [105]. It was proven that both methods failed to provide host-guest CD–ERY complexes, because of limited space within the CD inner cavity [106]. It was hypothesized that a packed 3D structure was formed via Van der Waals forces between the two components. The complexes displayed: extended stability in aqueous solution, increased bactericidal activity against *S. aureus*, inhibitory effects on osteoclastogenesis and good osteoblastic viability and differentiation. All these findings were considered as good premises for their application in the treatment of periprosthetic inflammations.

Nevertheless, the development of guest–host CD–ERY complexes was claimed by other groups of authors [107]. Using three different means of preparation (kneading, co-precipitation and freeze drying) and subjecting the final products to spectroscopic and thermal investigations, the authors suggested that inclusion was favoured by the co-precipitation and freeze-drying procedures. An aspect that should be mentioned is that in comparison to the earlier-described solvent evaporation technique, the co-precipitation method requires for the solvent evaporation to be performed at 50 °C in a drying stove. In addition, the freeze-drying method implicates mixing ERY and CD in water (in the absence of alcohol) followed by lyophilization. The investigation of ERY–CD antibacterial activity against *S. aureus* showed superior results compared to those of neat ERY, with the highest activity being attained by ERY–CD complexes prepared through co-precipitation [20,108].

An interesting route towards obtaining ERY–CD complexes was accomplished by replacing the β–CD with sugar grafted cyclodextrins. The background of this idea was to use the sugar molecule as chemoattractant, considering that many sugars such as manose (MAN) and glucose (GLU) are valuable carbon sources for bacteria and can readily permeate into bacterial cells through membrane sugar transporters [109]. The authors envisioned that the grafted sugar CD nanocarrier will act as a “Trojan horse” for ERY, thus favouring its cell uptake. ERY complexes with β–CD, MAN–CD and GLU–CD were prepared by a slightly modified kneading method. The final products showed an increase in the average diameter, good solubility, and no changes in the zeta potential, compared to blank modified CDs, therefore pointing towards a successful complexation. However, there is no evidence to prove the successful inclusion of ERY into the CD cavity. The complexes displayed enhanced in vitro delivery and accumulation of ERY in Gram-positive (*S. aureus*) and Gram-negative (*E. coli*, *P. aeruginosa, A. baumannii*) bacterial cells, leading to the reduction of MIC levels by a factor ranging from 3 to >100, compared to free ERY. Moreover, these complexes were able to prevent the development of bacterial resistance, indicating that sugar-modified CDs are excellent antibiotic carriers, capable of resolving the issue of multidrug-resistant bacteria.

Additionally, the same authors investigated the activity of these ERY complexes in the prevention and eradication of *P. aeruginosa* and *S. aureus* biofilms [110]. They reported a higher extent of penetration and uptake into *P. aeruginosa* biofilms compared to *S. aureus*, attributed to a more efficient interaction with the sugar residues. Considering the elevated antibacterial and antibiofilm activity, these complexes were proposed as a clinical alternative to catheter removal in the case of medical devices-related biofilm infections.

Similarly, successful preparation of ERY complexes via the solvent evaporation method was claimed when β–CD was replaced with a lactide modified β–CD [111]. It was stated that the modified β–CD has a higher polarity and consequently higher ability to bind ERY. The complexation improved ERY solubility, while in vitro investigations demonstrated a sustained release of 75% ERY over 12 h.

In the same line of thought, β-cyclodextrin modified with a variety of alkylamino substituents proved a synergic effect with ERY against Gram-positive and Gram-negative bacteria, significantly decreasing its MIC [112]. The modified CD induced ERY activity against Gram-negative pathogens. The synergic effect was attributed to the ability of modified CD to disrupt the outer membrane barrier of bacteria, thus favouring ERY access. These results indicate the alkylamino-modified CD as good potentiators or enhancers of ERY to combat drug-resistant pathogens.

An interesting study reported the preparation of CD-loaded ERY with a complex design aiming towards locally targeted ERY release to infection sites [113]. To this aim, ERY was modified with adamantane via a hydrazone pH-sensitive bond, which is cleavable in the slightly acidic medium of the infected sites. Adamantane was selected due to its ability to interact with ERY forming more hydrophobic compounds with higher stability and ability to bind to tissues more readily, increasing the tissue residence time and consequently the antibiotic effect. CD has been crosslinked with hexamethylene diisocyanate to give an insoluble matrix which was soaked in the modified ERY solution. Analysis of the in vitro release curves confirmed a faster ERY release in acidic media (pH = 5) compared to media of physiologic pH (7.4) and indicated high levels of drug loading. The microbiological investigations revealed that the CD/modified ERY formulations and neat ERY presented comparable in vitro antibacterial activities against *S. aureus* along with similar abilities of penetrating and killing the majority of bacteria in a mature biofilm within 7 h.

As a concluding remark, considering that some modelling studies indicate ERY inclusion in α–CD and β–CD less probable, it is easier to believe that non-guest–host ERY–CD complexes are attained. This did not minimize the improvement of ERY properties when combined with CD, especially with modified CD.

## 5. Gels

ERY gels for topical use are already commercial pharmaceutical products, prescribed for the treatment of skin infections, especially those caused by *P. acnes* [114]. The formulation has an official monograph in the United States Pharmacopeia, entitled “Erythromycin Topical Gel” and a monograph in combination with benzoyl peroxide. Recently, Erygel^®^ was registered and approved by the FDA for the treatment of *acne vulgaris* [115].

Nevertheless, the rapid increase in the development of antibiotic resistance, along with the challenges brought by poor drug penetration into the skin, encourages continuous research to unveil new directions within this field of therapy. In order to address these issues, combination therapy has been considered by scientists as an important tool in acne treatment and in overall drug delivery and tissue engineering. When it comes to dermatocosmetic applications, the gel base is an appropriate vehicle to increase the drug’s residence time on the skin, compared to other formulations such as alcohol-based solutions, while avoiding the oleaginous aspect of ointments or other common side effects such as dryness, irritation and redness. Moreover, the hydrogels exhibit pseudo-plastic properties with appreciable firmness, work of shear, stickiness and work of adhesion. They have a porous morphology which makes them suitable matrixes for tissue regeneration. Thus, gels incorporating different formulations of ERY (e.g., vesicles) or combinations of ERY and other bioactive compounds were tested as possible routes towards innovative formulations. Some of the findings covering this topic are presented below.

### 5.1. Gels Encapsulating Neat ERY

First, the effective ERY content in external administration forms for acne was proven to be at least 2%; as a result, nearly all tested gel formulations possessed this concentration [116]. Conventionally, ERY gels are prepared by dispersing the drug into a hydrogel which has established its suitability for dermal applications by surpassing clinical trials. Some traditional representatives are gels based on hydroxyethylcellulose or Carbopol resins [117]. However, other polymers, particularly those of natural origin, are considered to be promising gelling agents due to the improvements they bring about.

It was demonstrated that ERY has the potential to increase the intrinsic antimicrobial activity of the benzoyl peroxide gel through a possible catalysis of radical formation [118]. Furthermore, it was revealed that the ERY/benzoyl peroxide gel (Benzamycin^®^) presented stability issues, which could be surmounted by replacing the gel base (carbomer homopolymer), used in the original formula, with ***hydroxyethylcellulose*** [119]. By comparison, the addition of ***Carbopol 940*** yielded formulations with large variations of ERY content, because of its precipitation and aggregation [120]. The hydroxyethylcellulose gels incorporating dispersed ERY proved an outstandingly good stability, with 90% of the product’s initial activity being preserved following 1 month storage at 25 °C [121].

Adding ***polyamidoamine dendrimers*** (generations G2 and G3) to ERY–hydroxyethylcellulose hydrogels demonstrated an improvement in ERY release properties and a slight increase in the antibacterial activity against *S. aureus ATCC 29213*, *E. faecalis ATCC 2912* and *S. aureus* (clinical strain) [122].

ERY has also been encapsulated into a ***calcium polyphosphate*** hydrogel in order to create formulations targeted for the prevention and treatment of orthopaedic infections [123]. It was demonstrated that between ERY and the polyphosphate chains, there existed interactions, therefore retarding the drug release. A sustained release over two months has been achieved by compacting the semidried gel via a mechanical compressor.

***Pluronic F127*** hydrogels containing ERY were prepared by crosslinking a Pluronic F-127 diacrylate macromer under low-intensity UV light [62]. Briefly, micelles previously prepared from Pluronic monomer and ERY were mixed with a photoinitiator and irradiated with UV light for different amounts of times, in order to provide various crosslinking densities. It was observed that the ERY release rate and the antimicrobial activity against *E. coli* and *S. aureus* were dictated by the crosslinking density: a smaller degree of crosslinking induced higher antimicrobial activity. Additionally, the hydrogels allowed for relatively good L929 cell viability, proving their potential for use in the prevention of postoperative infections.

More intricate ERY hydrogels constructed with ***pluronic* *F127*** were prepared by using a combination including either *pectin* or *gelatine* and *glycerol* as a plasticizer. By varying the pluronic/gelatine, pluronic/pectin ratios, a series of hydrogels were prepared, which were further moulded to obtain films. The hydrogel films released ERY in a controlled manner, exhibiting antibacterial activity against *S. aureus*, while being non-toxic to human skin fibroblast, indicating potential as an antibacterial wound dressing [124].

***Poly(vinylalcohol****)* (PVA) in combination with various additives, such as PEG-600, glycerin, carboxymethylcellulose, caprylic/capric triglyceride oil and polyoxyethylene sorbitan mono-oleate (Tween80) were tested as ERY matrixes, claiming to be used in order to develop transdermal drug delivery systems [125]. However, ERY has not yet been established as a suitable drug for transdermal delivery, its topical use in acne treatment being a local, dermal one. Of the tested substances, it was shown that glycerin and PEG were more suitable to this end, due to their good miscibility with PVA and ERY, enabling good film-forming characteristics. Furthermore, glycerin favoured the drug release, in a sustained manner, slightly improved compared to the blank PVA matrix.

### 5.2. Gels Incorporating ERY and Co-Bioactive Ingredients

Aiming towards dermatocosmetic applications, hydroxyethylcellulose gels encapsulating the bioactive compound ***magnolol*** along with ERY were prepared and investigated. It was found that magnolol endowed the gels with thermosensitivity, and as a result changed their behaviour from liquid-like at 20 °C, to gel-like at 37 °C [126]. Even more, the presence of magnolol improved ERY release and granted great stability to the gels.

A fascinating study indicated ***honey*** as a valuable co-bioactive agent for developing ERY wound dressings. The study focused on examining the physico-chemical properties of hydrogel films prepared from PVA/ERY/honey and PVA/ERY/sucrose, by thermal annealing [127]. It was demonstrated that the addition of honey or sucrose improved the hydrogel’s bioadhesivity, and significantly enhanced the in vitro antimicrobial activity against *S. aureus* and *P. aeruginosa*, especially in the case of honey. In vivo wound healing investigation on rats showed that the presence of honey promoted early healing and had a positive influence on fibroblast proliferation and re-epithelialization, indicating a synergic effect of ERY and honey [128].

A similar study of the same authors presented the generation of hydrogel films analogous to the ones previously described, except for the fact that PVA crosslinking was performed with ***borax*** [129]. This allowed for the encapsulation of large amounts of ***honey***, which elevated the antibacterial activity, cell viability and proliferation. In addition, crosslinking with borax induced good mechanical properties, prevented hydrolytic degradation, and allowed for a better control over ERY release.

With the desire to develop biomaterials for tissue regeneration, ***hydroxyapatite*** (HAp) and ***collagen*** (Col) were used as bioactive agents in order to stimulate adhesion, proliferation and differentiation of osteoblastic cells [130]. Thus, PVA/HAp/Col gels embedding ERY were prepared by physical crosslinking via freeze–thaw treatment. The hydrogels presented good mechanical properties, swelling ability and antimicrobial activity, in relation to the crosslinking degree. Moreover, they induced enhanced MC3T3 cell growth compared to the untreated cell reference, indicating their ability to provide a bone-like matrix that could promote cell adhesion, proliferation and differentiation. It was envisaged that these hydrogels could be useful as sustained-release ERY carriers, with potential to enhance osteointegration and prevent prosthesis infection.

The same group published another study, focusing yet again on the production of bactericidal bone scaffolds with osteointegration potential. As a result, they designed ***strontium-doped calcium polyphosphate***/ERY/PVA hydrogels, by PVA-coating of an ERY-impregnated strontium-doped calcium polyphosphate scaffold via the slurry dipping method [131]. The authors pointed out that PVA coating increased the hydrogel’s elasticity and induced the drug release in a more controlled manner, facilitating the inhibition of *S. aureus* growth. Further, the formulation provided good cell viability on MC3T3-E1 cells and osteoclast inhibition in a murine RAW 264.7 macrophage cell line.

Intending to design medicinal products for the treatment of chronic inflammatory diseases, such as those encountered in periodontology, various ***antioxidants* *(krill oil***, ***aloe*** and ***aspirin***) were encapsulated along with ERY into *chitosan-based* hydrogels, as dual-action drug delivery systems [132]. The principal idea was to combine the antimicrobial activity of ERY with the antioxidant properties of krill oil, aloe or aspirin, along with the latter’s anti-inflammatory action, and bio-adhesive properties of chitosan, to formulate hydrogels for improving intradental administration of therapeutic agents within periodontal diseases mediated by ROS mechanisms. The hydrogels were prepared in two steps: (i) dispersion of ERY and antioxidants (single antioxidant or mixtures) into glycerol and glacial acetic acid, followed by (ii) incorporation of the mixture into a chitosan solution. This procedure led to a fine dispersion of ERY into the hydrogels, attributed to the occurrence of physical interactions. In vitro drug release investigations (PBS of pH = 6.8) demonstrated that ERY was delivered over a prolonged period, lacking any sign of burst effect and reaching 70% cumulative release in 10 h.

All samples showed in vitro antibacterial activity against *S. aureus*, even higher than the standards set by the European Committee on Antimicrobial Susceptibility Testing (Basel, Switzerland) for ERY sensitivity towards *S. aureus.* It was noted that hydrogels encapsulating both ERY and an antioxidant agent (aloe or aloe and krill oil or aspirin and krill oil) showed significantly larger inhibition zones (31.6, 31.8, and 33.1 mm) compared to those incorporating only ERY (25.2 mm), therefore suggesting a synergic effect. In addition, the hydrogels containing antioxidant agents showed a general increase in shear bond strength to dentin after 24 h and a good stability, maintaining the antioxidant effect over a period of one month, while the investigation was carried out. All these findings indicated the addition of antioxidant agents to ERY hydrogels as a practical pathway for the development of drug release systems for dentistry use purposes.

Combining ERY with ***zinc*** into hydrogels is still a popular research topic in view of application for acne treatment. *Sayyafan* et al. initiated a clinical trial on two topical gels based on hydroxyethyl cellulose containing either (i) 2% ERY or (ii) 2% ERY and 1.2% zinc acetate, in order to validate their synergic effect in the treatment of mild to moderate inflammatory *acne vulgaris* [133]. The double-blind study, carried out on 103 patients, showed that both gels significantly diminished comedons, papules and pustules lesions, being more efficient compared to hydroalcoholic solutions containing the same concentrations of bioactive compounds (ERY, or ERY + Zn). However, the ERY gel proved to be less effective than the Zineryt^®^ lotion [134]. The treatment based on the ERY/Zn gel was more effective than the ERY gel, with respect to reducing the number of acne lesions and the grade of acne severity, indicating that zinc acetate increased ERY’s inhibitory effect on *Propionibacterium acne* growth. Nevertheless, the differences were not significant between the two therapeutic strategies.

### 5.3. Gels Incorporating ERY Vesicles

***ERY niosomes*** (LD~90%, 4.5 µm) were encapsulated into a gel based on *crosslinked polyacrylic acid* polymer (Carbopol 934) in order to enhance ERY skin penetration and retention [59]. It was remarked that the encapsulation of niosomes into the gel significantly improved their stability, the drug leakage being less than 5% when the gel was refrigerated and less than 50% when it was kept at room temperature, while the naked niosomes showed more than 30% and 80% ERY leakage in similar conditions. This stability improvement was attributed to the fact that the viscous gel prevented niosomes fusion. The diffusion and retention of ERY into the skin were investigated ex vivo in a Franz diffusion cell by applying the gel formulations on human cadaver skin, revealing superior results for the niosomal gel: 41% ERY retention into the skin after 24 h and only 21% for the plain ERY gel. These results were justified by the ability of niosomes to form ERY reservoirs within the skin tissue, promoting its prolonged release and slower diffusion. Nevertheless, no clear evidence of the niosomes’ presence in the skin was provided.

A somehow similar study reported the encapsulation of ***ERY-SLNs*** (~176 nm) into a Carbopol gel [75]. The resulting products exhibited good properties for topical applications such as homogeneity, spreadability and extrudability. The in vitro release tests (PBS of pH = 6.4) showed no significant variations in the ERY release profile between the SLN gel and a reference ERY gel, reaching 90.94% and 87.94% cumulative release, respectively. The antimicrobial activity against *S. aureaus* displayed a slightly more efficient activity of ERY gel after 18 h, but a significantly more efficient activity of the SLN gel after 24 and 30 h, pointing towards a prolonged release and an improved therapeutic effect.

Cubosomal gels were prepared by dispersing ***ERY cubosomes*** in Carbopol, selecting various ratios [67]. Even though no significant alterations were observed in the ERY release kinetics from cubosomal gel compared to pristine ERY cubosomes, the cubosomal gel showed antibacterial activity against *Saccharomyces cerevisiae*, while no such activity was observed for ERY cubosomes. This could be the result of the long period of incubation during the antimicrobial tests (3–5 days) and higher fluidity of the cubosomes, which favoured their faster consumption, allowing the bacteria to grow. Therefore, these findings denote a sustained ERY release from the cubosomal gel due to the lower fluidity.

As presented in Section 3.2, ***ERY Janus micromotors*** are a research hot spot of the recent years, with promising results for targeted drug delivery. Nevertheless, naked micromotors may be disrupted or passivated in a physiological environment, resulting in premature leakage of the drug or limited self-propulsion performance. To overcome this shortage, ERY micromotors were encapsulated in situ into hydrogels based on chitosan, formerly crosslinked with dialdehyde starch via Schiff-base bonds [103]. The hydrogel’s components are natural originating polymers, biocompatible and biodegradable, crosslinked via a covalent reversible Schiff base, where the reaction equilibrium can be shifted towards the reagents under environmental stimulus, such as pH [135]. Thus, in a physiological environment, the hydrogel will slowly degrade, controlling the ERY micromotor’s release. This way, the encapsulation into hydrogels reduces external influences on the micromotors and improves their selective release effect. The hydrogels were thixotropic, having the ability to be injected through a 26-gauge needle which is widely used in biomedical applications. In vitro drug release experiments confirmed that gel encapsulation retarded the ERY release, and moreover indicated a pH-dependent release profile, a more rapid release being remarked in acidic pH (6.5), in line with the pH sensibility of the Schiff bonds which promoted a faster hydrogel degradation [136,137,138,139]. This is envisioned as a positive aspect, considering that infection sites have a slightly more acidic pH [140]. Antibacterial tests on *E. coli* and *S. aureus* displayed significantly larger inhibition zones for the hydrogels loaded with ERY micromotors compared to similar hydrogels loaded with ERY capsules (without Pt) or neat ERY.

***ERY microemulsion***-based hydrogel was prepared using isopropyl myristate oil and Tween 80 surfactant to first prepare the ERY microemulsion which was then stabilized into Carbopol and triethanolamine to form the gel [141]. Along with ERY, isotretinoin was encapsulated, in order to investigate their synergism. ERY vesicles were around 79 nm, and their Zeta potential was around 5 mV. The ex vivo tests of drug permeation on mice skin showed good retention of both drugs, while in vitro studies on *P. acnes* showed enhanced efficacy compared to plain ERY, indicating a synergic activity of two antibiotics, the gel formulation being recommended for acne treatment.

## 6. Fibres

Fibres belong to a group of materials that are highly suitable for drug delivery system applications, particularly those directed at wound healing [142,143]. In this context, ERY has been employed in such materials, which were developed from a variety of natural or synthetic polymers. Some of the formulations which make the subject of this chapter are reviewed below.

With the purpose of designing drug release systems with a controlled delivery in the intestines, ERY composite fibres were prepared using ***hydroxypropyl methylcellulose phthalate*** (HPMCP), a natural originating polymer which is soluble at a pH of 5.5 or higher [144]. The fibres were prepared by electrospinning a blend solution of HPMCP/ERY (9/1, *w*/*w*) of various concentrations. The inclusion of ERY was observed (i) to improve the electrospinning process by increasing the solution’s conductivity and (ii) to change the fibre’s morphology, from a cylindrical shape to a more ribbon-like one. By varying the concentration of the solution, fibres with an average width ranging from 0.2 to 1 μm were attained. The pH-sensitive nature of HPMCP endowed the fibres with the ability to exhibit a pH responsive release of ERY, noting that in artificial intestinal juice (pH = 6.8 at which HPMCP is soluble), the release was 2.5x times faster compared to that in artificial gastric juice (pH < 5.5). Another substantial remark which emerged from the investigations was that the ERY release rate could be controlled by modifying the fibre’s width, thinner fibres releasing the drug faster than thicker ones, which is in alignment with the fact that larger surface-to-volume ratios improve the diffusion of the drug. Thus, the fibres presented slow and little release in gastric pH (around 4.6% in 2 h), whilst nearly all ERY was released at an intestinal pH, confirming that HPMCP can be used as an ERY matrix for a successful intestinal delivery.

Aiming to achieve a synergistic antibacterial effect by combining ERY and ZnO nanoparticles in view of wound treatment, nanocomposite fibres were electrospun from a blend solution of ***sodium carboxymethyl cellulose***/***PVA*** (1/1) with ERY (5%, *w*/*w*) and ZnO nanoparticles (3%, *w*/*w*) [145]. In order to improve the hydrolytic stability, the fibres were further crosslinked with glutaraldehyde. SEM investigations revealed continuous randomly aligned nanofibres with no bead defects, and with an average diameter of 234 nm, while in vitro release tests displayed a prolonged ERY release over 30 days. Fibres encapsulating both ERY and ZnO nanoparticles revealed high fibroblast viability and synergic inhibition effects on relevant pathogens, such as *S. aureus* and *E. coli*, which are valuable findings, with regard to their wound healing applications.

Core-shell fibres with a more complex composition, including ERY-loaded ***poly(caprolactone)*** as the fibre core, and zein-containing TiO_2_ as the outer shell, were prepared by coaxial electrospinning [146]. The selection of the core-shell design was justified by the need for improving the controlled release properties, while each component was chosen in view of reaching fibre multifunctionality, by combining their valuable intrinsic properties. Thus, poly(caprolactone) was selected for its ability to function as a drug delivery matrix and its potential to act as a scaffold for tissue regeneration, zein was introduced due to its capacity to improve cell attachment and proliferation, and TiO_2_ nanoparticles for their antifungal activity and their capability to ease the electrospinning process [147]. The average diameter of the completed fibres was approximately 625 nm, with a clear delimitation of the core-shell structure, and an even distribution of the TiO_2_ nanoparticles. In vitro ERY release tests in PBS revealed a burst release of 21% within the first 5 min, followed by a sustained release, as follows: 46% after 1 h, 90% after 25 h, and 98% after 72 h. It was stated that ERY encapsulation into the core-shell fibres retarded the drug’s release, due to a slower penetration of the medium into the fibre core. The antimicrobial activity against *E. coli*, *S. aureus*, *B. cereus*, and *Salmonella*, measured after 24 h of incubation, was slightly lower compared to that of neat ERY, possibly as a result of the slower ERY release. Nevertheless, it was expected that the formulations could ensure a good physical barrier against pathogens when applied on wounds.

With the same goal of application in tissue regeneration, ERY has been encapsulated in both core and shell components of coaxial ***poly(caprolactone)***/**PLGA-PVA** fibres [148]. It was demonstrated that ERY improved the electrospinning process and promoted an increased porosity and surface wettability of the fibres. ERY encapsulation into fibres promoted a prolonged release higher than 4 weeks, strong inhibition of *S. aureus*, and the growth and differentiation of rat bone marrow stem cells, being envisaged as potential matrices for implants.

A similar study investigated the possibility of a synergic effect between ERY and tretinoin, when encapsulated into ***poly(caprolactone*)** fibres, starting from the idea that tretinoin enhances ERY’s penetration through the skin (Figure 14). The foundation of this study was to attain a nanofibre material with increased antibacterial efficiency, suitable for topical applications as an anti-acne patch [149].

From the available information, it is not very clear how the fibres encapsulating both ERY and tretinoin were prepared, nor what physico-chemical characteristics they possessed, for the reason that the paper focused most of its attention on the tretinoin-encapsulating fibres. Nevertheless, a comparative analysis was provided of the antibacterial activity against two *S. aureus* strains between the fibres embedding ERY and tretinoin/ERY, respectively, suggesting the absence of any synergistic effect.

Recent papers prove once again the interest within research groups for the use of ***chitosan*** in order to prepare ERY fibre formulations for topical skin delivery, either as a polymer matrix or for preparing ERY nanoparticles which were further encapsulated into fibres. The justification for selecting chitosan as a formula component is mainly explained by its bioadhesivity, a property required for topical application of nanofibres.

Fibres based on ***chitosan*** and ***PVA***, encapsulating lidocaine hydrochloride and ERY gelatine nanoparticles (ERY NPs), were prepared with the purpose of evaluating the dual drug release behaviour [95]. Fibres presenting no defects were successfully prepared by electrospinning of a blend solution of the components, with a PVA/chitosan mass ratio of 96/4, followed by crosslinking with glutaraldehyde. While the ERY content was clearly specified (10% ERY NP (*w*/*w*)), no explicit information regarding the content of lidocaine was provided. However, lidocaine-containing fibres were also prepared and characterized via a similar procedure, leaving room for a proper interpretation of the influence of ERY. It appeared that the ERY NPs induced a lower swelling ratio of the fibres, a truly surprising aspect considering the high hydrophilicity of gelatine. Lidocaine was released more rapidly than ERY, recording 84% cumulative release after 72 h, compared to 75% for ERY. The antimicrobial tests against both *S. aureus* and *P. aeruginosa* demonstrated good activity for the fibres embedding ERY NPs, even though their activity was lower in contrast to that of neat ERY NPs. Nevertheless, the successful combination between the ERY antibiotic and the local anaesthetic lidocaine into a fibrous material accompanied by their prolonged release over a period of 72 h points to its use as a wound healing dressing with great potential for pain alleviation and infection prevention.

ERY-loaded chitosan nanoparticles were encapsulated into ***cellulose acetate nanofibres***, following a two-step procedure: (i) obtaining ERY-loaded chitosan nanoparticles (ERY NPs) by ionic gelation followed by (ii) electrospinning of a blend dispersion of ERY NP (5, 8, 12, 15% wt%) in cellulose acetate [98]. This technique led to smooth homogeneous fibres with an average diameter around 141 nm, a high-water holding ability, no cytotoxic effect on normal human fibroblasts and a capacity to inhibit the growth of both Gram-positive and Gram-negative bacteria, indicating them to have good potential as dressings for the healing of infected wounds. It was concluded that the fibres containing ERY NPs presented a significantly superior antibacterial activity compared to that of fibres encapsulating neat ERY, due to the higher activity of ERY-NPs compared to neat ERY.

## 7. ERY Loading into Mesoporous Oxides

Mesoporous nanoparticles are suitable materials for drug delivery applications due to their large surface area, which can accommodate generous amounts of bioactive principles, therefore being excellent carriers within the development of drug delivery systems and implantable local-delivery devices [150].

***Mesoporous silica nanoparticles***, under their renowned forms, namely MCM-41, MCM-48 and SBA-15, are considered to be potential candidates for drug delivery, since they have adequate properties for in vivo applications, such as biocompatibility, non-toxicity, and degradability in the biological milieu [151]. Nevertheless, in order to obtain a controlled release, it is necessary to functionalize the pore walls, with the purpose of decreasing their size and increasing the anchoring of the encapsulated compound, preventing its leakage.

In this line of thought, ***SBA-15 mesoporous silica***, which has unidirectional mesoporous channels with diameters of 6–10 nm, was functionalized with octyltrimethoxysilane (C8) and octadecyltrimethoxysilane (C18) aiming to investigate the influence of the aliphatic chain length on ERY release, choosing calcined SBA15 as a reference [152]. It was envisioned that the aliphatic chains will promote ERY anchoring through hydrophobic forces and decrease the silica wettability in aqueous solutions, retarding the release. The drug loading optimization led to the highest loading efficiency of 34% for calcined SBA-15 and 13% and 18% for the functionalized SBA-15 silicas. The functionalization with C18 favoured the encapsulation of a higher amount of ERY and retarded its release by a factor of nearly one order of magnitude compared to bare SBA-15, justified by the stronger ERY anchoring due to hydrophobic interactions. Moreover, C18-modified SBA-15 displayed a much slower release of ERY, even slower than that of MCM-14 mesoporous silica, which has significantly smaller pores (3.8 nm), clearly highlighting the key role played by the hydrophobic forces involved in ERY anchoring into the pores.

The same group also investigated the influence of C8 and C18 functionalization of ***MCM-48 mesoporous silica*** with pore sizes of 3.6 and 5.7 nm [153]. Initially, one of the first remarks of the study was that the pore size was connected to the ERY release rate, smaller pores favouring a slower release. The drug release rate was further decreased by functionalization of MCM-48, especially with C18, supporting once more the hypothesis of strong ERY anchoring via hydrophobic forces.

In a similar study, ***SBA-15 mesoporous silica*** was functionalized with 12-tungstophosphoric acid (TPA) and the role of functionalization on ERY release was investigated [154]. The authors arrived at the same conclusion, that functionalization retarded the drug release, possibly due to the binding of the macrolide to TPA.

A more complex design was reported by *Pourjavadi and Tehrani*, who developed core shell formulations by coating ERY-loaded ***mesoporous* *silica nanoparticles*** (MSN) functionalized with PEGylated chitosan [155]. The design took into account the benefits of the two polymers: the naturally originating chitosan is a pH-sensitive biopolymer while PEG induces colloidal stability and improves the blood circulation lifetime. In order to ensure a good bonding of the PEGylated chitosan to the MSN, they were previously functionalized with 3-aminopropyl triethoxysilane. The preparation of this formulation was carried out in two steps: (i) loading ERY into the functionalized MSN by sonication in acetonitrile, followed by (ii) coating with PEGylated chitosan under stirring in solution (pH = 6). This procedure led to uniform nanospheres with diameters ranging from 100 to 150 nm, covered with a thin PEGylated chitosan layer. A lower drug loading was noted for the coated nanoparticles vs. uncoated ones (12% and 6%, vs. 18%) attributed to ERY leakage during the coating process. However, the coating was beneficial for achieving a prolonged release. While the uncoated nanoparticles released ~95.5% ERY during the first 6 h (pH = 7.4), the coated ERY particles released only 35.5% ERY in similar conditions, and the drug release increased to 65.4% after 24 h. Additionally, coating with PEGylated chitosan endowed the MSN particles with pH sensitivity, showing 62.3% and 96.2% ERY release in acidic medium (pH = 5.5), in 6 and 24 h, respectively. The more rapid release rate in acidic medium was explained by the protonation of chitosan’s free amine groups, weakening the hydrogen bonds between MSN and chitosan, collapsing the shell and promoting drug release. The authors assumed that these results show a great potential of chitosan-coated ERY-functionalized MSN formulations for oral administration with controlled release in gastric acidic medium. However, no indication about the behaviour of MSN in any of the investigated pH values was provided.

Another intricate formulation was produced by loading the drug in ***graphene quantum dots/mesoporous silica***, with the intent of creating a photodynamic wound dressing [156]. A thorough investigation conducted in light of the targeted application revealed that these composite materials displayed good hemocompatibility, ability to produce ROS under light irradiation and exhibited antimicrobial activity against both Gram-positive and Gram-negative bacteria (*E. coli*, *S. aureus*), which was significantly improved by irradiation. In vivo studies of the wound healing properties on mice revealed superior therapeutic effects of the composites compared to the neat components, by virtue of the synergic effect of quantum dots and ERY.

Another mesoporous material, investigated for its controlled properties for delivering ERY, was based on *titanium zirconium oxide* [157]. ***Mesoporous titanium zirconium oxide nanospheres*** of ∼360 nm diameter, with uniform pore diameters ∼3.7 nm and variable Ti to Zr ratios, were loaded with ERY from hexane solution, after which they were separated by centrifugation (Figure 15).

A higher loading capacity was observed in connection with the increase in titanium content, while the release profile in PBS (pH = 7.4) remained similar, presenting a burst effect in the first 24 h followed by a sustained release. These inorganic mesoporous nanoparticles demonstrated good hydrolytic stability in PBS for 21 days, lack of degradation under physiological conditions, and good biocompatibility with SaOS2 osteoblast-like cells (MIC = 250 mg/L), therefore proving their potential to be used as a local drug delivery agent for the treatment of bone-related complications.

Loading of ERY into ***mesoporous ZnO nanoparticles*** was also reported, claiming that the drug release was controlled by UV irradiation, the authors recommending the formulation as a smart antiseptic [158].

## 8. Metallic-Based Nanoparticles as ERY Carriers/Cofactors

Magnetic nanoparticles are considered to be a significant doorway towards the development of novel drug delivery systems. They can function as drug vectors for the targeted release of chemotherapeutics, but also as cofactors for improving the antibacterial activity of antibiotics [159]. ERY formulations with magnetic nanoparticles were reported as excellent candidates for drug delivery systems with enhanced antibacterial activity, even against resistant strains.

ERY formulations with improved antimicrobial activity were developed by preparing ***magnetite nanoparticles*** functionalized with sulfanilic acid and chitosan [160]. This design was conceived in order to achieve an improved antimicrobial activity and a suitable biocompatibility, by combining the complementary properties of each component, that is: (i) the ability of magnetic nanoparticles to act as vectors for improving the aspects of drug delivery and controlled release; (ii) the ability of *sulfanilic acid* to ensure a good dispersion of magnetite nanoparticles, thus preventing their agglomeration; and (iii) the biocompatibility and antimicrobial activity of chitosan, which also brings positive charges on the nanoparticle surface, beneficial for cellular adhesion and retention at the target site, thus allowing for the modulation of in vitro drug release. Therefore, the complementarity of these particular characteristics was expected to increase the antimicrobial activity of ERY, leading to a high efficiency at lower therapeutic dosages and consequently lower side effects. The preparation of this complex was achieved in situ, by mixing all the components, followed by washing and crosslinking with glutaraldehyde. TGA analysis indicated chitosan as being predominant, and the content of ERY, magnetic nanoparticles and sulfanilic acid was 1.76%, 3.26%, and 3%. This elaborated formulation revealed significantly lower MIC values against *S. aureus* and *E. coli*, indicating an increase in ERY’s antimicrobial activity up to 25 times.

By coupling ERY with PEGylated iron oxide nanoparticles via iron alkoxide, ERY carrying nanovehicles were prepared, which exhibited remarkably improved antimicrobial activity against *S. pneumonias*, as shown by the decreased MIC values from 0.25 to 0.12 μg/mL [161]. The superior activity has been hypothesized to be induced by the affinity of this bacterial strain for iron.

Erythromycin-capped ***gold nanoparticles*** (ERY-Au(0)NPs) of ~9.2 nm were prepared by a one-pot method consisting of heating a mixture of ERY/gold chloride in NaOH solution [162]. It was proposed that capping was accomplished through the coordinative bonds between gold and OH and CO units of ERY. The antimicrobial activity assay against *S. aureus ATCC 2985* and *Salmonella typhi ATCC 6539* indicated significantly higher activity compared to pure components (ERY and gold chloride), highlighting the role of the nanoparticles or even the occurrence of a synergistic effect.

Other investigations reported that the antibacterial activity of ERY against *P. aeruginosa* can be significantly improved when combined with ***calcium phosphate nanoparticles*** [163] or ***CuO nanoparticles*** [164], whereas its combination with ***silver nanoparticles*** showed increased activity against *B. cereus*, *Bacillus subtilis* (*B. subtilis*), *Klebsiella pneumonia* and *Vibrio cholerae* [165], MDR bacteria *S. aureus* strains (methicillin-resistant and methicillin-susceptible), *Proteus mirabilis* [166], *E. coli*, *K. pneumoniae* and *P. aeruginosa* [167,168].

## 9. Overview of Current Progress in ERY Formulations

Despite the substantial work on the ERY formulations, only few of them were in vivo investigated, and even fewer have reached clinical trials. Table 1 summarises the stage of the investigation for the main ERY formulations.

## 10. Quantification of ERY

A most important aspect in the development of ERY formulations is the ability to quantify the amount of drug which is available for delivery. Therefore, a series of parameters have been used in order to measure this capability, such as the **encapsulation efficiency** (EE) and **drug loading capacity** (abbreviated: DLC/DL/LD/LC). However, ERY brings with itself challenges in this matter, lacking a quantification method which is at the same time accessible and reliable. Moreover, a quantification method is also needed for release profile studies, biocompatibility studies, etc.; therefore, this lack greatly impacts the entirety of the antibiotic’s formulas characterization.

The EE represents a percentage expressed through the amount of incorporated drug identified in the formula, in relation to the initial concentration used to produce the formulation. Therefore, EE is characterized by the following expression, where W_t_ is the total amount of drug present in the formula, while W_i_ is the total quantity of drug added initially during preparation [170]:EE% = (W_t_/W_i_) × 100%(1)

The drug-loading capacity is defined as the ratio between the amount of entrapped drug and the total weight of the carrier (e.g., lipid, surfactant, etc.), with the following formula, where W_i_ represents the initial amount of drug added, W_r_ represents the residual amount of drug or the amount of free, unencapsulated drug, and W_c_ represents the weight of the carrier [77]:LC% = (W_i_ − W_r_)/W_c_ × 100%(2)

However, no matter which one of these two parameters is chosen to be calculated, or if the authors include both in their studies, the determination of drug content is mandatory in both cases and can be performed by a series of methods.

### 10.1. Ultraviolet Spectroscopy

The first method of assay brought into discussion is the spectrophotometric determination of ERY by UV–VIS spectroscopy, the most frequently used analytical means of quantification. Although it is extensively described in the literature, it must not be omitted that ERY exhibits a disturbingly weak absorption band at 285 nm, creating difficulties in the measurement of small drug quantities, which is most often the case for laboratory-level drug formulation designs. Nevertheless, some papers reported the use of UV spectroscopy by measuring the neat ERY absorbance as a method for drug quantification.

As a result, some methods aiming to increase the UV absorption of ERY have been portrayed, one of which involves ***acid hydrolysis***. The procedure was first developed and optimized in 1953, shortly after the drug’s discovery, due to the observation of a yellow coloration following the treatment of ERY with a solution of hydrochloric acid (6 N), which led to an absorption maximum at 485 nm. The authors performed the reaction over a water bath heated at 50 °C for 30 min, later replacing the hydrochloric acid with sulfuric acid in order to avoid corroding the UV–VIS spectrophotometer. The procedure was optimized by means of sulfuric acid concentration, time and temperature effects on colour development, in order to determine the optimum parameters for colour formation. As the colour intensity was similar for 12 and 14 N acids, sulfuric acid 27 N was used as a reagent, taking into consideration that a concentration around 13 N is reached by adding it to an equal volume of ERY aqueous solution in PBS. It was revealed that the intensity increased approximately 30% compared to the use of hydrochloric acid 6 N. Moreover, it was concluded that temperature did not have a significant effect over the reaction. As a downside, the study found that the determination method is not specific for ERY, its degradation products interfering with the measurement [171].

Two years later, another study stated that after strong acid hydrolysis at elevated temperatures, ERY exhibits maxima at 226, 267, and 485 nm, with the only band which obeys Beer’s law being the one at 226 nm. However, other degradation products of ERY also present absorption in the 226 nm range and limit the usefulness of acid hydrolysis as an assay method, according to some authors [172].

For this method, a series of acids are currently used within the scientific community; however, the three most often employed are H_2_SO_4_, HCl and H_3_PO_4_. Lacking a standardization of the method, many authors have taken the liberty of varying the acid concentration, acid-to-solvent ratio, reaction time and other parameters, therefore leaving room to question the veracity of the results, due to a lack of reproducibility.

One study describes the procedure by using acetonitrile and water (1:1) along with H_2_SO_4_ [173]. The authors used a temperature of 50 °C and a time of 30 min to ensure the completion of the reaction, explaining that the yellow coloration which appears is due to the reaction between ERY’s sugar moiety and the strong acid. Dilutions were made in the range of 5–50 ppm, selecting the detection wavelength at 480 nm.

Another method designed to increase ERY’s UV absorption is based on using ***alkali reagents along with strong acids***, as follows [174]. First, prior to the determination, the standard, the alkali reagent and the buffer solutions are prepared. The phosphate buffer with a pH of 7.0 is prepared by the dissolution of anhydrous KH_2_PO_4_ and K_2_HPO_4_ in distilled water, and the alkali reagent is obtained from Na_3_PO_4_ and NaOH in distilled water by heating. For the determination, two standard solution aliquots and two blank aliquots are necessary, the latter being treated with H_2_SO_4_ (0.5 N) each, and left to rest at room temperature for approximately 1 h. Additionally, purified water is added to the standard solutions. After time has passed, NaOH (0.1 N) is added to the blanks and mixed, followed by the addition of the alkali reagents to all samples, which are mixed and subjected to heating at 60 °C for 15 min. The cooling should be performed rapidly on an ice bath, until reaching room temperature, followed by dilution with distilled water. The absorbance should be read at a wavelength of 236 nm against distilled water, subtracting the blank values from the standard ones. The blank aliquot treated with H_2_SO_4_ produces the cyclic degradation product, anhydroerythromycin. The alkaline treatment produces the development of an unsaturated ketone (9-keto-10-ene), with its maximum absorbance being a shoulder, found at 236 nm.

Another way to improve ERY’s absorbance consists of using ***bromocresol purple*** in the form of its ion pair dye complex which interacts with the desosamine fraction of ERY in an acidic buffer (pH = 1.2). This determination, also known as the colorimetric method, is not specific, due to the fact that the reaction takes place with all tertiary amines. Furthermore, it is frequently used to determine concentrations of 250 µg ERY/mL in tablets [174].

A similar method also targets the desosamine component of ERY, this time using ***p-dimethylaminobenzaldehyde***. However, it is not specific for ERY, but it is linear in a concentration interval of about 10 to 35 µg/mL [174].

This being said, Table 2 comprises the results reported for ERY quantification from various formulations along with the parameters used for determination method. As it can be observed, a series of different maximum absorption wavelengths are currently being chosen, with notable variations. A significant remark is that some studies only use UV–VIS to determine the release profile of their formulation, and not the drug content. Another significant remark is the fact that directly analysing formulations encapsulating ERY is questionable in terms of accuracy, due to the fact that particles, e.g., liposomes, tend to aggregate, leaving room to wonder if the Lambert–Beer law (which applies to diluted media) is valid in this case.

### 10.2. High-Pressure Liquid Chromatography

***High-pressure liquid chromatography*** is an analytical chromatographic technique which is generally used for the quantification and separation of components in a mixture. This method is widely used in the pharmaceutical industry in order to quantify antibiotics in pharmaceutical products. The method also provides information about the qualitative composition of drug samples. Due to its precision and speed, HPLC is a much more attractive technique than the classical microbiological assay for the quantification of antibiotics in both pharmaceutical products and body fluids. Therefore, this analytical method has mostly replaced microbiological assays, with the only downside being the high cost of the method [176].

A study comparing the drug concentrations obtained by microbiological and chemical assays in body fluids revealed only minor differences between the two, with slightly higher values in the case of the microbiological determinations. However, a strong correlation was discovered amongst the two, concluding that the bioassay reflected a high accuracy [177,178].

Table 3 presents some examples of the HPLC method used to quantify ERY in the literature, most commonly the mobile phase being a mixture of acetonitrile and phosphates in various proportions.

Another chromatographic method applied to determine ERY concentration is by ***liquid chromatography coupled with mass spectrometry*** (LC-MS). LC-MS is an analytical determination which has the benefit of synergically combining the separation abilities of HPLC with the mass analysis properties of ***mass spectrometry*****.** Therefore, the determination is superior to HPLC.

A study which focused on preparing ERY magneto-liposomes quantified the antibiotic by LC-MS, using diphenhydramine as an internal standard, and two 0.1% formic acid solutions, one in water and one in acetonitrile as mobile phase. The results showed a drug-loading efficiency of 87.9 ± 2.2% in the case of ERY–liposomes and 97.8 ± 1.5% for ERY–magneto–liposomes [49].

### 10.3. Thermogravimetry

Thermogravimetric analysis is an analytical method built on the principle of recording changes in weight in relation to increasing temperatures. The method requires a precision balance and a furnace that is programmed for a linear rise in temperature with time [179]. A study which focused on preparing magnetic nanostructures used TGA analysis to estimate the amount of ERY, by calculating the amount of antibiotic as a difference between the weight loss of the formulation containing ERY and the neat one, lacking ERY [160].

Another study mentioned the use of TGA in order to determine the amount of ERY absorbed into mesoporous materials. The determination was based on the fact that mesoporous inorganic materials are thermally stable, while ERY decomposes around 220–450 °C [180]. Thus, the TGA method proves to be a simple and rapid determination technique of organic substances from organic/inorganic composites [152]. Moreover, the TGA method was also used for the determination of ERY adsorbed into mesoporous silica nanoparticles coated with chitosan, or chitosan–PEG copolymers. To discriminate between different organic compounds (ERY or chitosan), TGA curves were recorded for different formulations (mesoporous silica coated with polymer and mesoporous silica encapsulating ERY and coated with polymer), calculating the ERY amount by subtracting the mass loss of the two formulations (Figure 16).

### 10.4. Near Infrared Spectroscopy

An interesting means of quantifying active ingredients is that by near infrared (NIR) spectroscopy, even though FTIR it is not recognized as being a quantitative determination method, NIR however, can be used for quantitative determinations of drug concentrations, due to its increased sensitivity. The method is based on measuring signals of molecular vibrations in the infrared range, especially the asymmetric vibrations, i.e., stretch vibrations involving hydrogen bonds (C-H, O-H, N-H).

Therefore, this method is gaining more and more attention within the pharmaceutical industry for quality control. One study aimed to evaluate the applicability of NIR spectrometry to determine the active substance concentration in several dermal pharmaceutical preparations, amongst which ERY was also subjected as sample. However, the development of a quantitative determination model for ERY was unsuccessful, due to the fact that ERY is composed of multiple subgroups of variable solubility, depending on its concentration, thus creating a difficulty in obtaining accuracy from this analytical technique. The study determined that an increase in water proportions determined a decrease in accuracy in all cases, therefore leading to the conclusion that the method is concentration-dependent. However, there is room for more investigations to be performed in this field of study [181].

### 10.5. Microbiological Assay

When it comes to pharmaceutical substances with antibacterial activity, such as antibiotics, the microbiological determinations are a key characterization method in order to determine the specific activity and potency of the drug or formulation. However, this assay has also been used for the quantification of antibiotics. Furthermore, an antibiotic’s antibacterial activity per milligram of product can be determined by comparing the inhibition zones of the sample with those of standards of identical concentrations. A standard curve is made, using values given by appropriate dilutions of both the sample and stock solution, followed by calculating the activity of the tested antibiotic. This method relies on the concentration-dependent modifications of antibacterial activity of antibiotics over reference bacteria, producing a linear response along with two parameters, the y-intercept which equivalates with the concentration and the slope of the curve which translates into the drug’s potency [182].

Microbiological determinations are simple, accurate and inexpensive methods which offer results comparable to those obtained by HPLC. However, each method has its own benefits, the microbiological assay being the only accepted standard method for estimating antibiotics activity loss. All of the microbiological procedures include either the diffusion of the drug in agar or its dilution in agar or broth [183].

The ***Kirby–Bauer disk diffusion*** method consists in developing a pathogenic microorganism on Mueller–Hinton agar and adding various antibiotic-impregnated paper disks. Depending on the diameter of the inhibition zones, if any, the activity of an antibiotic against the tested bacterial specie can be determined.

Another similar method, the ***broth dilution***, is based on subjecting the isolate to various concentrations of the antibiotic in a broth medium. This technique can be performed either by microdilution testing, with the possibility of using a Microtiter plate, or by macrodilution testing which is carried out in standard test tubes.

The ***agar dilution method*** implies incorporating the antibiotic into the solid agar medium, inoculating the surface of the Petri dish with a standard bacterial concentration, thus providing a specific MIC value for the tested antibacterial agent.

Furthermore, the ***agar diffusion method*** also known as the cylinder-plate or cup-plate technique estimates antibiotic bioactivity by the diffusion of a drug through the agar via a vertical cylinder. The microorganism growth is restricted into a circular region surrounding the cylinder with antibiotic. This technique gives insight on the inhibition zones and on the antibiotic dosage. However, the correlation between the diameter of the inhibition zones and the drug concentration, from a solution, has only been considered theoretically [184].

Therefore, all methods which are used to evaluate antibiotics from a microbiological susceptibility point of view have the MIC (minimum inhibitory concentration) value as the common denominator, which is considered to be the standard parameter used to characterize antibiotics by their lowest necessary concentration which is able to inhibit visible bacterial growth, following a night of incubation [183].

Table 4 summarizes the EE/LC values reported in the literature as being determined by microbiologic assay. In order to apply this method correctly, with the purpose of quantification, a well-established protocol must be in place. Moreover, the difference between an antibiotic’s activity and its concentration should be emphasized, due to the fact that microbiological activity can variate, in such a way that the same amount or concentration of the same antibiotic from two different samples could possibly have different antibacterial activities. Therefore, the correlation between activity or potency and concentration represents a key step in correctly evaluating drug concentrations in various formulations. One of the reasons for this is the fact that many variables intervene in the process of evaluating an antibiotic’s activity, such as the chaotic, unpredictable manner of bacterial resistance development, degradation of the drug to various degrees, etc.

To conclude, all these methods of quantification indicate that EE% and LC% cannot be compared amongst different studies due to various methods implied. Even if the same method of quantification is used, being that by UV–VIS spectroscopy or bioassay, within the method, there can exist slightly significant differences such as wavelength, reaction conditions, bacterial strains or microbiological determination methods, etc., making it difficult to find the relevance in comparing two values obtained by different means. Another important aspect, which is less discussed, is represented by the fact that due to a lack of a standard terminology and guidelines set by competent authorities, the authors have taken liberty to modify the terms used to describe ERY quantification, therefore leading to a series of different terms describing drug loading and encapsulation. Inevitably, confusion arises from the distinction between terms representing the same parameter. For example, the terms “entrapment” and “encapsulation” are currently used interchangeably, while terms such as “percentage drug entrapment”, “drug loading efficiency”, “loading contents”, “yield of encapsulation”, and “drug loading capacity” can also be found [187].

## 11. Conclusions

Since its discovery in 1952, ERY was employed for inhibiting a large variety of Gram-positive and -negative strains both internally and externally. However, the major drawbacks, especially the low water solubility, drug resistance, and side effects, have stimulated research for increasing the therapeutic efficiency of ERY. Due to their already proven advantages, (nano)carriers are a promising solution for development of ERY formulations with improved therapeutic efficacy. Therefore, the encapsulation of ERY in (nano)carriers is necessary in order to overcome the antibiotic resistance issue, to achieve higher bioavailability and minimize side effects. As the review of the available literature showed, important steps were undertaken to achieve this goal. ERY encapsulated in gels was reported well before the 1990s, but the drug resistance matter was not resolved. This drawback was overcome with the liposomal ERY, the first studies being reported early in the 1990s. As the research in lipid-based (nano)carriers progressed, a larger pallet of carriers became available for transporting ERY, which brought additional advantages for ERY-based therapies. The literature revealed that ERY was encapsulated in almost all known drug delivery formulations, such as various vesicles (liposomes, ethosomes, niosomes, micelles, cubosomes, solid lipid nano(micro) particles), micro(nano)spheres, inclusion complexes, gels, fibres, films, as well inorganic carries, that is, mesoporous oxides and metallic-based nanoparticles, of which preparations and performances were reviewed in this paper. However, except for the traditional ERY gels used for the treatment of acne, only few studies were finalized with clinical trials or at least in vivo tests. It can be seen that the research on ERY formulations is still in an early stage, in which various materials and conditions were explored in order to control the particle size, loading degree, drug release, and synergistic therapeutic effect in the presence of other bioactive agents, etc. In essence, all the data summarized in this review should be a useful tool for scientists working in ERY drug formulation and other hydrophobic drugs. An important impairment in the development of new ERY formulations appears to be the difficult quantification. To come to the aid of scientists working on ERY or other hydrophobic drugs, a section was dedicated to the methods and strategies for ERY quantification.

To conclude, it appears that among the ERY formulations reviewed in this paper, the vesicles are extremely promising, as they can be applied “as such”, or can be further used to develop more complex formulations, by their dispersing in gels, fibres or films.

Seeing how much focus is currently invested in the development of nanomedicine and advanced drug delivery formulations, there can only be room left to wonder why it is that so few of the developed formulations have reached the state of clinical studies or further clinical applications.

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
