# Peer review of "Erythromycin Formulations—A Journey to Advanced Drug Delivery"

_pharmaceutics, 2022, doi:10.3390/pharmaceutics14102180_

Round 1
Reviewer 1 Report
RE: Erythromycin formulations – a journey to advanced therapeutic use.
The above manuscript title was sent to me for review. The manuscript is a review of the past work on erythromycin formulations.
My Comments/Observations:
I have gone through the work severally and I found out that the work was well selected and appropriate references were cited and referenced. The indebt of the review was commendable. The authors have also made a serious effort to place the figures at the relevance section of the work.
However, the following few gray areas were observed and need to be corrected
a. 20oC, to gel-like at 37oC [125]. – page 24 and in many areas the degree was not properly written.
b. (Tween80) instead of (Tween 80) such was found in some areas where reference were made to polymers
c. Trade mark symbol with superscript® R was not well placed in many of the sections where reference was made.
In the conclusion section of the work:
I suggest the author should look at the merit in other formulations and expand his conclusion on the best formulations with high advantages in terms of release, sustainability, efficacy and stability.
I found out too there are other formulations with very promising in addition to the one the author has pinned down.
Recommendation/The overall rating: The work is a very fine and well written review and I recommended that except another issue arises, it should be accepted for publication.
Author Response
We thank to the reviewer for her/his effort to read our manuscript and to do constructive observations meant to improve its content for the readers’ benefit. Your expertise and kindness are highly appreciated. We carefully read again the manuscript and implemented all the suggested corrections, as follows.
Reviewer comment
I have gone through the work severally and I found out that the work was well selected and appropriate references were cited and referenced. The indebt of the review was commendable. The authors have also made a serious effort to place the figures at the relevance section of the work.
However, the following few gray areas were observed and need to be corrected
- 20oC, to gel-like at 37oC[125]. – page 24 and in many areas the degree was not properly written.
- (Tween80) instead of (Tween 80)such was found in some areas where reference were made to polymers
- Trade mark symbol with superscript®R was not well placed in many of the sections where reference was made.
Authors answer
We have gone through the manuscript trying to fix all the spelling/grammatical errors, including those highlighted by the reviewer.
Reviewer comment
In the conclusion section of the work:
I suggest the author should look at the merit in other formulations and expand his conclusion on the best formulations with high advantages in terms of release, sustainability, efficacy and stability.
I found out too there are other formulations with very promising in addition to the one the author has pinned down.
Authors answer
The Conclusions section was expanded to comprise all the formulations discussed in the review. Also a table including the stage of exploration of their performances (in vitro, in vivo, clinical) has been provided in order to better highlight the results reported for ERY (nano)carriers.
Once again we thank to the reviewer for her/his constructive feed-back on our manuscript. We are seeking for the reviewer opinion regarding all the corrections made in the manuscript.

Reviewer 2 Report
The authors wrote the manuscript ID pharmaceutics-1951293 titled Erythromycin formulations – a journey to advanced therapeutic use. The manuscript reviewed comprehensively the novel drug delivery systems for erythromycin where the authors discussed the different delivery systems, their method of preparation, tools of characterisation with special emphasis on entrapment efficiency and drug loading. The authors also discussed the method of quantifications used in these novel drug delivery systems. The references are adequate, up to date and relevant to the topic as well as the gap in gathering all erythromycin novel drug delivery systems is achieved. An addition merit of this review is the gathering of all novel drug delivery systems of erythromycin regardless of their route of administration. The conclusion support the citations and the future perspective of the authors was included .
As in the era of antimicrobial resistance this review is very important to grab the attention of the researchers and regulatory affairs for repurposing the use of antibiotic through different delivery system to avoid drug resistance.
I do recommend the manuscript to be published after minor corrections.
· The title is interesting but I think “… advanced therapeutic use” imply usage of ERY rather than reflecting advanced drug formulation of ERY which is the core subject of the review. I would suggest if the title can be modified to reflect the advanced drug delivery of ERY.
· As I mentioned earlier one of the merit of this review is the presence of different drug delivery systems of erythromycin. It will be very useful and informative if the authors could add a summary table containing the ERY formulation, route of administration, in vitro and in vivo studies (If any), clinical trials (If any) and clinical applications.
· As the author focus on the method used to quantify ERY I would suggest to delete section 9.6 since there is no report or information about using NMR to quantify ERY.
· I would suggest to the authors to move section 9.2 to the end as it is the only method of quantification includes the bacteria for quantification.
· Scheme 1 spelling of “ciclodextrin complexis” need to be revised.
Author Response
We thank to the reviewer for her/his effort to read our manuscript and to do constructive observations meant to improve its content for the benefit of readers. Your expertise and kindness are highly appreciated. We carefully read again the manuscript and improve it in accordance with the reviewer comments, as follows.
Reviewer comment
The authors wrote the manuscript ID pharmaceutics-1951293 titled Erythromycin formulations – a journey to advanced therapeutic use. The manuscript reviewed comprehensively the novel drug delivery systems for erythromycin where the authors discussed the different delivery systems, their method of preparation, tools of characterisation with special emphasis on entrapment efficiency and drug loading. The authors also discussed the method of quantifications used in these novel drug delivery systems. The references are adequate, up to date and relevant to the topic as well as the gap in gathering all erythromycin novel drug delivery systems is achieved. An addition merit of this review is the gathering of all novel drug delivery systems of erythromycin regardless of their route of administration. The conclusion support the citations and the future perspective of the authors was included .
As in the era of antimicrobial resistance this review is very important to grab the attention of the researchers and regulatory affairs for repurposing the use of antibiotic through different delivery system to avoid drug resistance.
I do recommend the manuscript to be published after minor corrections.
- The title is interesting but I think “… advanced therapeutic use” imply usage of ERY rather than reflecting advanced drug formulation of ERY which is the core subject of the review. I would suggest if the title can be modified to reflect the advanced drug delivery of ERY.
Authors answer
The title has been changed.
Reviewer comment
- As I mentioned earlier one of the merit of this review is the presence of different drug delivery systems of erythromycin. It will be very useful and informative if the authors could add a summary table containing the ERY formulation, route of administration, in vitroand in vivo studies (If any), clinical trials (If any) and clinical applications.
Authors answer
A table has been made and provided in the manuscript (see table 1).
Reviewer comment
- As the author focus on the method used to quantify ERY I would suggest to delete section 9.6 since there is no report or information about using NMR to quantify ERY.
- I would suggest to the authors to move section 9.2 to the end as it is the only method of quantification includes the bacteria for quantification.
- Scheme 1 spelling of “ciclodextrin complexis” need to be revised.
Authors answer
All these suggested corrections were done. Besides, a figure reflecting the way of using TGA for ERY quantification has been provided (Figure 16).
Once again we thank to the reviewer for her/his kind feed back on our manuscript. We are seeking for the reviewer opinion regarding all the corrections made in the manuscript.

Reviewer 3 Report
Comments:
1) This is a wonderful review article describing a comprehensive examination of the strategies for Erythromycin formulations, characterization, and therapeutic uses. One suggestion to the authors, is to put a cautionary note in the end, if possible, not just for this review but also for all papers on nanomedicine or advanced formulations asking why to have all those seemingly, potentially exciting Erythromycin formulations not resulted in any clinical benefit. This is an important issue that all of us need to think about. and the authors may comment on this in their manuscript.
2) Fig 6: The bar for group B is missing in fig 6, which represents >75% reduction. Please include it in the figure.
3) Remove the underline in mass spectrometry in section 9.3.
4) Define the acronyms and bracket them. if authors are using it for the first time in the text. e.g., MIC values…See line... P. acnes at concentrations below MIC values [14].
Please check all the acronyms throughout the draft and define them, if you are using them for the first time.
5) Line… The latter-mentioned limitation was one of the driving forces for the recent development of novel targeted ERY formulations.
Replace (for) with (in) …….in novel targeted ERY formulations.
6) Considering that the development of new antibiotics has reached a plateau, another concerning issue that makes the design of improved ERY formulations a current priority….
Remove “a current priority” from the text or just make it easy to follow.
7) Under section 2.1 liposomes, Line: .. [37].
Decreasing the amount of cholesterol using the DRV method increases the E.E (51%) as well as a dramatic increase in liposomal size. No explanation for attaining such a high size for liposomes was given. What is your own opinion about the larger size of liposomes? Try to give your opinion and put in the review if possible e.g., The large size might be due to the fact of ……... (Add a reason)
For me, it is due to lipid–cholesterol interactions that induce the condensation effect. Higher cholesterol can induce more condensation effects. As a result, smaller liposome size and vice versa. Please see this article and put as a reference: de Meyer, Frédérick, and Berend Smit. "Effect of cholesterol on the structure of a phospholipid bilayer." Proceedings of the National Academy of Sciences 106.10 (2009): 3654-3658.
8). There are a bunch of long and complex sentences which is distracting sometimes. It would be great to make them split into short sentences and make it easy for everyone to follow.
Author Response
We thank to the reviewer for her/his effort to read our manuscript and to do constructive observations meant to improve its content for the readers’ benefit. Your expertise and kindness are highly appreciated. We carefully read again the manuscript and correspondingly improve it, as follows.
Reviewer comment
- This is a wonderful review article describing a comprehensive examination of the strategies for Erythromycin formulations, characterization, and therapeutic uses. One suggestion to the authors, is to put a cautionary note in the end, if possible, not just for this review but also for all papers on nanomedicine or advanced formulations asking why to have all those seemingly, potentially exciting Erythromycin formulations not resulted in any clinical benefit. This is an important issue that all of us need to think about. and the authors may comment on this in their manuscript.
Author answer
We totally agree with the reviewer about the necesity of such a comment. A paragraph has been introduced in Conclusions to highlight this aspect.
Reviewer comment
- Fig 6: The bar for group B is missing in fig 6, which represents >75% reduction. Please include it in the figure.
Author answer
The bar for the group B in figure 6, is missiing for „The Excellent response percentage” because it was reported to be 0%. A note has been added in the figure caption to reflect this.
Reviewer comment
- Remove the underline in mass spectrometry in section 9.3.
4) Define the acronyms and bracket them. if authors are using it for the first time in the text. e.g., MIC values…See line... P. acnes at concentrations below MIC values [14].
Please check all the acronyms throughout the draft and define them, if you are using them for the first time.
5) Line… The latter-mentioned limitation was one of the driving forces for the recent development of novel targeted ERY formulations.
Replace (for) with (in) …….in novel targeted ERY formulations.
6) Considering that the development of new antibiotics has reached a plateau, another concerning issue that makes the design of improved ERY formulations a current priority….
Remove “a current priority” from the text or just make it easy to follow.
Author answer
The suggested corrections were done.
Reviewer comment
7) Under section 2.1 liposomes, Line: .. [37].
Decreasing the amount of cholesterol using the DRV method increases the E.E (51%) as well as a dramatic increase in liposomal size. No explanation for attaining such a high size for liposomes was given. What is your own opinion about the larger size of liposomes? Try to give your opinion and put in the review if possible e.g., The large size might be due to the fact of ……... (Add a reason)
For me, it is due to lipid–cholesterol interactions that induce the condensation effect. Higher cholesterol can induce more condensation effects. As a result, smaller liposome size and vice versa. Please see this article and put as a reference: de Meyer, Frédérick, and Berend Smit. "Effect of cholesterol on the structure of a phospholipid bilayer." Proceedings of the National Academy of Sciences 106.10 (2009): 3654-3658.
Author answer
We agree with the reviewer that the lipid–cholesterol interactions can be the driving force for the lyposomes size. The manuscript has been changed to reflect this and the suggested reference has been provided.
Reviewer comment
8). There are a bunch of long and complex sentences which is distracting sometimes. It would be great to make them split into short sentences and make it easy for everyone to follow.
Author answer
The English language has been revised for clarity.
Once again we thank to the reviewer for her/his kind feed back on our manuscript. We are seeking for the reviewer opinion regarding all the corrections made in the manuscript.
